# On Reward Functions for Self-Improving General-Purpose Reasoning

## Abstract

Prompting a Large Language Model (LLM) to output Chain-of-Thought (CoT) reasoning improves performance on complex problem-solving tasks. Moreover, several popular approaches exist to "self-improve" the CoT reasoning abilities of LLMs on tasks where supervised (question, answer) datasets are already available. An emerging line of work explores whether self-improvement is possible without these supervised datasets, instead utilizing the same large, unstructured text corpora as used during pre-training. This would overcome the data availability bottleneck present in current self-improvement methods, and open the door towards *compute-only scaling* of language model reasoning ability. We investigate a fundamental question in this line of work: What constitutes a suitable reward function for learning to reason during general language model pretraining? We outline the desirable qualities of such a reward function and empirically demonstrate how different functions affect what reasoning is learnt and where reasoning is rewarded. Using these insights, we introduce a novel reward function called Reasoning Advantage (RA) that facilitates self-improving CoT reasoning on free-form question-answering (QA) data, where answers are unstructured and difficult to verify. We also perform an exploratory experiment optimizing RA on general unstructured text using offline RL, and our analysis indicates that future work should investigate methods for generating a more diverse set of CoTs.

## 1 Introduction

Large Language Models (LLMs) have become increasingly effective at solving complex reasoning tasks (Huang & Chang, 2022; Kojima et al., 2023; Wei et al., 2023; Havrilla et al., 2024b). A key driver of this success has been the discovery of Chain-of-Thought (CoT) reasoning (Wei et al., 2023), whereby a model outputs a step-by-step "thought process" before arriving at a final answer.

While some CoT reasoning ability emerges naturally from pretraining on unstructured web-text data (Fu et al., 2023), it is through further supervised finetuning (SFT) on curated question-answering (QA) datasets (Saparov & He, 2023), as well as Reinforcement Learning from Human Feedback (RLHF) (Ouyang et al., 2022), that CoT becomes such a powerful tool. Considerable effort is being placed in curating large-scale (question, CoT, answer) datasets (Cobbe et al., 2021; Saparov & He, 2023; Liu et al., 2023), with models increasingly being used "in the loop" to help generate initial reasoning traces or refine existing ones (Zelikman et al., 2022; Zhang et al., 2024). In certain domains like mathematics, it is also possible to further automate dataset generation by sampling many CoTs and selecting those which lead to ground-truth answers (Zelikman et al., 2022). However, despite these recent advancements, there are significant limitations to relying on curated datasets for improving CoT abilities. It is becoming increasingly difficult and prohibitively expensive to curate sufficiently challenging, large-scale (question, CoT, answer) datasets across the diverse set domains that today's general models can tackle. For instance, a popular benchmark of just 500 graduate-level biology, physics, and chemistry questions with CoT reasoning and answers cost over $120,000 to produce and required thousands of human expert hours (Rein et al., 2023).

To address these limitations, an emerging line of work explores self-improving CoT reasoning ability in a self-supervised setting—leveraging the large, unstructured datasets used for pretraining (Zelikman et al., 2024) instead of relying on curated QA or RLHF datasets. In this new setting, the LLM learns to produce CoT reasoning for the task of next-token prediction: **given $n$ tokens from the pre-**

**training corpus, the model generates a CoT and receives a reward based on how well the CoT helps predict the following $m$ tokens.** This is an exciting prospect, as we have trillions of tokens of unstructured text encompassing much of human knowledge. Therefore, learning to self-improve CoT reasoning on pretraining scale data might overcome the data availability bottleneck in current self-improvement methods, opening the door towards *compute-only scaling* of reasoning ability.

While there have been some initial efforts towards self-improving CoT during pretraining, we investigate a fundamental problem in this emerging line of work: **What constitutes a suitable reward function for reasoning during general language model pretraining?** In Section 4, we outline the desirable qualities of such a reward function, and in Section 5.1, we empirically investigate how different functions affect:

1. *What* reasoning is rewarded—the ability to distinguish effective CoT reasoning

2. *Where* reasoning is rewarded—the ability to pick useful locations to produce CoT reasoning

To our knowledge, our work is the first to provide this type of analysis on reward functions towards self-improving CoT reasoning on unstructured text. Our investigations reveal critical shortcomings in commonly used reward functions, including an inability to differentiate between meaningful CoT reasoning and random word sequences (poor *what*: failing to reward effective reasoning), as well as a tendency to incentivize reasoning at locations where predicting following tokens is trivial (poor *where*: inability to pick out useful locations for reasoning). Drawing on these insights, we introduce a novel reward function called Reasoning Advantage (RA), an augmentation of standard language modeling loss, and show that it addresses many of these limitations.

To facilitate more efficient study of self-improving CoT reasoning, we also introduce an open-ended, free-form QA dataset called MMLU-FREE-FORM by adapting the popular MMLU dataset (Hendrycks et al., 2020) to be closer to the unstructured text setting. Specifically, by removing its multiple-choice format and requiring models to generate full, unstructured answers—which are hard to verify using exact-match accuracy heuristics (see Figure 6). Our purpose in creating MMLU-FREE-FORM is to make the smallest possible change to MMLU that reveals the limitations of existing reward functions. It acts as an intermediate benchmark between improving CoT reasoning using curated (question, CoT, answer) datasets and the challenging, unsolved task of self-improving CoT reasoning on unstructured text.

MMLU-FREE-FORM does not allow for using exact-match accuracy as a reward metric (similar to unstructured pretraining text), and yet offers a higher density of clear opportunities for CoT reasoning compared to typical pre-training corpora. This makes it an ideal stepping-stone towards the ultimate goal of self-improving CoT reasoning on unstructured text. In Section 5.2, we demonstrate that RA is the only reward function which enables self-improvement of CoT reasoning on MMLU-FREE-FORM, improving zero-shot transfer accuracy on GSM8K (Cobbe et al., 2021) by nearly 7%, compared to barely when trained with other reward functions.

Using our Reasoning Advantage (RA) reward function, we conduct an initial experiment on self-improving CoT reasoning on general unstructured text using OpenWebMath (Paster et al., 2023), a collection of 14.7 billion tokens of maths-heavy text. Our results in Section 6 indicate that the offline RL algorithm employed is not sufficiently powerful to escape the local optimum of extremely conservative CoT reasoning that just summarizes previous information instead of trying to solve the problem. Future work should investigate methods for generative a more diverse set of CoTS. To facilitate future work, we will open-source all of our code, which runs on academic compute.

In summary, our main contributions are as follows:

- We establish desirable criteria of reward functions for self-improving CoT reasoning on unstructured text at pretraining scale.

- We provide empirical evidence demonstrating how different reward functions impact both the quality of CoT reasoning (*what* reasoning is rewarded) and the ability to pick out useful locations to produce CoT reasoning (*where* reasoning is rewarded).

- We introduce MMLU-FREE-FORM, an open-ended QA dataset that facilitates more efficient study of self-improving CoT reasoning and reveals the limitations of commonly used reward functions. It serves as an intermediate benchmark between curated QA datasets and general language modeling on unstructured text.

- We propose Reasoning Advantage (RA), a novel reward function based on clipped normalized loss, and demonstrate that RA is the only reward function which facilitates self-improvement of CoT reasoning on MMLU-FREE-FORM, a key step towards self-improving reasoning on unstructured, pretraining-scale text.

- While our work does not solve the challenging problem of self-improving CoT reasoning on unstructured text at the pretraining scale, we conduct an initial experiment and provide key insights into how future work might make further headway in this direction. Specifically, while we are unable to generalize when optimizing RA using a simple offline RL algorithm on OpenWebMath (Paster et al., 2023), our analysis suggests that future works should investigate ways to better explore the space of possible CoTs. This includes moving towards more online RL algorithms, in order to escape the local optimum of learning conservative CoT reasoning strategies that just summarize prior information from the context.

- We will open source all of our code, which runs on academic compute, to facilitate future work in this direction.

## 2 BACKGROUND

**CoT Reasoning**   Given $n$ prefix tokens $\mathbf{p}$, performing CoT reasoning refers to an LLM $\mathcal{M}$ generating a sequence of reasoning tokens $\mathbf{r}$ before the $m$ answer suffix tokens $\mathbf{s}$. The goal of generating CoT reasoning tokens before the final answer is to maximize $P_{\mathcal{M}}(\mathbf{s}|\mathbf{p}, \mathbf{r})$, the probability of the answer suffix tokens $\mathbf{s}$ conditioned on both the prefix $\mathbf{p}$ and the CoT reasoning tokens $\mathbf{r}$. The prefix-suffix pair can be any token sequence, ranging from question-answer pairs in mathematical datasets to arbitrarily split sentences from an unstructured text corpus.

Traditionally, CoT reasoning has been elicited by pretending few-shot examples of (question, CoT, answer) to the prefix. This approach relies the pattern-completion tendencies of LLMs to continue this structure for subsequent outputs. Alternatively, it has also become popular to elicit CoT reasoning by appending prompts like "Let's think step by step." to the prefix (e.g., to the end of input questions), especially for instruction-tuned models.

**Self-Improving CoT Reasoning as Reinforcement Learning**   Self-improvement refers to any process where an LLM is finetuned on self-generated data, leading to performance gains without human intervention or assistance from larger models. This process can be framed as a Reinforcement Learning (RL) problem. In RL, an agent interacts with an environment by taking actions $a \in A$ in states $s \in S$ to maximize cumulative rewards. The agent receives a reward $R_t = R(s_t, a_t)$ after each action $a_t$ and aims to learn a policy $\pi(a|s)$ that maximizes the expected cumulative discounted reward $G_t = \sum_{k=0}^{\infty} \gamma^k R_{t+k}$, where $\gamma \in [0, 1]$ is the discount factor.

In the context of CoT generation, each token can be viewed as an action $a_t$, with the current string of generated tokens representing the state $s_t$ so far. We focus on a sparse reward setting where rewards are 0 until CoT generation is complete, and with a discount factor $\gamma = 1$. The reward function maps the prefix $\mathbf{p}$, CoT reasoning tokens $\mathbf{r}$, and answer suffix $\mathbf{s}$ to a real number $R(\mathbf{p}, \mathbf{r}, \mathbf{s}) \in \mathbb{R}$, with higher rewards for CoTs that better predict the suffix. As long as this reward function doesn't require external intervention from humans or more powerful models, optimizing it through RL methods constitutes self-improving CoT reasoning.

**Self-Improving CoT Reasoning Using Supervised Datasets**   When a supervised dataset of (question, answer) pairs is available, accuracy can serve as a reward function:

$$R_{\mathrm{acc}}(\mathbf{p}, \mathbf{r}, \mathbf{s}) = \begin{cases} 1 & \text{if } \arg\max_{\mathbf{s}'} P_{\mathcal{M}}(\mathbf{s}'|\mathbf{p}, \mathbf{r}) = \mathbf{s} \\ 0 & \text{otherwise} \end{cases} \tag{1}$$

In this case, we can sample multiple CoTs and finetune on those that lead to correct answers (Dong et al., 2023; Zelikman et al., 2022). Iterating this process yields increasingly high-quality CoT generation, and this iterative self-improvement is equivalent to online reinforcement learning (Zelikman et al., 2022). There are also more complex methods, such as Process Reward Models (PRMs), which provide dense rewards for each step in a CoT and address credit assignment challenges (Ma et al., 2023; Wang et al., 2023; Havrilla et al., 2024b; Lightman et al., 2023).

**Self-Improving CoT Reasoning on General-Purpose, Unstructured Data** This setting explores the possibility of self-improving CoT reasoning given only an unstructured corpus of text, without access to a curated dataset of (question, CoT, answer) or (question, answer) pairs. In this setting, the model generates and inserts intermediate CoT reasoning at various points in a sequence of tokens (for example, at various points in a web-document that shows how to apply the quadratic formula).

A key challenge in this setting is evaluating the performance of CoT reasoning tokens inserted into general-purpose text. The accuracy-based reward $R_{acc}$ is ineffective here, as it would almost always be 0, providing minimal learning signal. Instead, language modelling performance—the log-likelihood of the suffix conditioned on the prefix and CoT—serves as a more natural starting point for a reward function:

$$R_{\text{loss}}(\mathbf{p}, \mathbf{r}, \mathbf{s}) = \log P_{\mathcal{M}}(\mathbf{s}|\mathbf{p}, \mathbf{r}) \tag{2}$$

We aim to help advance the field towards this setting, enabling self-improving CoT reasoning on unstructured text at the pretraining. In this paper, we specifically focus on identifying key shortcomings of commonly used reward functions and introducing a new function to address these limitations.

## 3 RELATED WORK

**LLM Reasoning** Various works have looked at improving the reasoning capabilities of LLMs. Rajani et al. (2019) improve the commonsense reasoning ability of language models by training on human explanations for commonsense problems. Nye et al. (2021) generate tokens in a "scratchpad" for intermediate computations when solving multi-step reasoning problems. On difficult algorithmic tasks, Pfau et al. (2024) show that LLMs can even be trained to leverage meaningless filler tokens under dense supervision, in place of legible CoTs. Further, theoretical analyses by Merrill & Sabharwal (2023) and Feng et al. (2024) show that CoT improves the expressivity of Transformers (Vaswani et al., 2017).

**LLM Self-Improvement Using Supervised Datasets** Iterated learning approaches involve LLMs generating new outputs and using "successful" ones to improve generation quality (Anthony et al., 2017; Vani et al., 2021; Polu et al., 2022). Such methods have been applied to LLMs (Zelikman et al., 2022; Huang et al., 2022; Chen et al., 2024). However, much of the research on LLM self-improvement has been limited to question-answer domains where accuracy is an appropriate success measure, such as multiple-choice questions or simple numeric answers. This limitation is evident in the policy gradient objective approximated by STaR (Zelikman et al., 2022): $\nabla J(M, X, Y) = \sum_i \mathbb{E}_{\hat{r}_i, \hat{y}_i \sim p_M(\cdot|x_i)}[\mathbb{1}(\hat{y}_i = y_i) \cdot \nabla \log p_M(\hat{y}_i, \hat{r}_i|x_i)]$, which makes use of an indicator function with respect to ground truth labels. Clearly, this breaks down in settings where ground truth labels are not available, such as open-ended or "free-form" QA setting as well as general-purpose language modelling. Havrilla et al. (2024a) show that Expert Iteration (Anthony et al., 2017), a self-improvement method based on iterative Supervised Fine-Tuning (SFT), outperforms RL in their evaluations. Building on this, our work extends RAFT (Dong et al., 2023), which also uses iterative SFT, by introducing a new reward function called Reasoning Advantage (RA) for filtering synthetically generated CoTs.

Process Reward Models (PRMs) (Ma et al., 2023; Wang et al., 2023; Havrilla et al., 2024b; Lightman et al., 2023) have been used to enhance reasoning via Reinforcement Learning (RL) by rewarding individual problem-solving steps in a CoT. However, PRM training is computationally expensive, usually involving backtracking and resampling from specific points in the CoT, and these points from which to resample are usually determined by hard-coded heuristics such as new line breaks.

**Self-Supervised LLM Self-Improvement** Quiet-STaR (Zelikman et al., 2024) looks to self-improve reasoning during general language modeling. Zelikman et al. generate a CoT at *every* token in an unstructured text document, using the negative cross-entropy loss on the suffix tokens as a reward. They employ REINFORCE (Williams, 1992) to optimize the loss of the suffix $\mathbf{s}$ given a prefix $\mathbf{p}$ and a reasoning trace $\mathbf{r}$, with a baseline for variance reduction. Importantly, performing CoT reasoning at every token is highly computationally expensive, making it difficult to use for pretraining-scale datasets and also limiting the length of CoT sequences that can be learnt (the reasoning learnt in Quiet-STaR is quite short and simple). Regardless, Quiet-STaR provides key insights into *how* to optimize for reward on general, unstructured text—a very difficult problem. Our work aims to take a step back and investigate the reward functions we optimize to self-improving

reasonin ability, with particular focus on *what* reasoning we should be rewarding and whether we can take steps towards determining *where* is the best place to produce CoT reasoning.

RHO-1 (Lin et al., 2024) investigates whether the sample efficiency of general language pretraining can be improved by selectively training on more useful tokens in a dataset, instead of training on all tokens. Lin et al. (2024) show that pretraining a model in this way enhances downstream reasoning ability, and we are excited for future work to investigate a combination of RHO-1 with our proposed RA reward function (i.e., to perform RL for CoT on datapoints that are suitable for reasoning, not noisy, and not yet learned).

## 4 REWARD FUNCTIONS FOR SELF-IMPROVING CoT REASONING

In Section 2, we framed self-improving Chain-of-Thought (CoT) reasoning as a Reinforcement Learning (RL) problem. Given $n$ tokens from a pre-training corpus (the prefix $\mathbf{p}$), the model generates a CoT $\mathbf{r}$ and receives a reward based on how well the CoT helps predict the following $m$ tokens (the suffix $\mathbf{s}$). Previous works have primarily explored two reward functions for self-improving CoT reasoning: loss and accuracy. Here, we explore other potential reward functions and their characteristics from the perspective of facilitating self-improving CoT reasoning on unstructured web-text at pretraining scale.

There are several key criteria to consider when designing such a reward function. Primarily, it should reward high-quality reasoning over CoTs containing logical errors or simply random characters. As shown in Section 5.1, this is not always the case. Moreover, for the purposes of *self-improving* CoT reasoning, the reward function must not depend on an stronger source of intelligence (i.e., using a more powerful LLM to verify the correctness of its CoT). Further, for reasonable use on pretraining scale datasets, evaluating the function should be fast and ideally parallelizable—requiring a minimal number of model forward passes.

In this work, we do not consider using an LLM-as-judge to evaluate or verify CoTs since: (1) it may rely on a stronger model, which is not self-improvement, and (2) while one could use the same model for both generation and verification, this approach incurs too much computational overhead to apply to pretraining scale data as it requires the decoding of an answer to be verified against the ground truth, and the verifier itself needs to generate CoT tokens. We also do not consider accuracy-based metrics, since free-form answers are often impossible to verify using exact-match, and using an LLM-as-judge to compute accuracy faces the issues mentioned above. Thus, we choose to focus on the family of "loss-based" reward functions. These functions compute the token-by-token log-likelihood of the suffix tokens $\mathbf{s}_{0,...,m-1}$, given the CoT $\mathbf{r}$ and prefix $\mathbf{p}$:

$$\begin{aligned}
\log P(\mathbf{s}|\mathbf{p},\mathbf{r}) = \ &\log P(\mathbf{s}_0|\mathbf{p},\mathbf{r}) \\
&+ \log P(\mathbf{s}_1|\mathbf{p},\mathbf{r},\mathbf{s}_0) \\
&+ \log P(\mathbf{s}_2|\mathbf{p},\mathbf{r},\mathbf{s}_{0,1}) \\
&+ ...
\end{aligned} \tag{3}$$

The most basic reward function in this family is $R(\mathbf{p},\mathbf{r},\mathbf{s}) = \log P(\mathbf{s}|\mathbf{p},\mathbf{r})$. This family of reward functions offers several key advantages. They are computationally efficient, since they can be evaluated by an autoregressive model in a single forward pass and can be parallelized across a batch of CoTs. Also, they do not require access to any external form of intelligence, a requirement for self-improvement. Most importantly, this family of functions does not rely on an using exact-match accuracy to compare with the answer suffix, enabling multiple valid answers and accommodating ambiguity in formatting (a key property of unstructured text).

While there are many possible ways to augment the basic loss-based reward function $R(\mathbf{p},\mathbf{r},\mathbf{s}) = \log P(\mathbf{s}|\mathbf{p},\mathbf{r})$, we focus our analysis on two key modifications: *clipping* the log probabilities and incorporating a *baseline* value.

**Clipping:** We clip (aka clamp) the minimum value of the token-level log probabilities to some $-\epsilon$ such that $R_{\text{clipped}}(\mathbf{p},\mathbf{r},\mathbf{s}) = \sum_{i=0}^{m} \max\left[\log P(\mathbf{s}_i|\mathbf{p},\mathbf{r},\mathbf{s}_{0:i}), -\epsilon\right]$. This constrains the loss contribution of each suffix token to the range $[-\epsilon, 0)$. In Section 5.1, we demonstrate that this clipping mechanism helps reward functions distinguish between well-formed CoTs containing a few logical errors and degenerate CoTs that resemble random tokens.

**Baseline Incorporation:** We explore incorporating a baseline value both with normalization ($R - B)/B$ and without normalization $R - B$, where $R$ is the reward and $B$ is the baseline value. A full

| Criteria | Accuracy | Loss | Loss with baseline | RA | LLM-as-judge |
|---|---|---|---|---|---|
| Uses no external intelligence | Yes | Yes | Yes | Yes | Yes[2] |
| Rewards good reasoning over random | Yes | **No** | **No** | Yes | Yes |
| Robust to multiple choices in answer | **No** | Yes | Yes | Yes | Yes[3] |
| Robust to answer perplexity | Yes | **No** | **No** | Yes | Yes |
| Fast and parrallelisable | No[1] | Yes | Yes | Yes | **No** |

Table 1: To what extent different reward functions meet our criteria. By RA, we mean loss augmented with clipping and the no CoT baseline, as defined in Appendix A. [1]whilst we do derive a generation free variant 'expected accuracy' in Appendix A that is as fast as loss based methods, the variant of accuracy used widely through the literature requires answers to be sampled, and so is slow. [2]Whilst acting as a verifier may be possible for larger models under heavy prompting, we found it difficult to consistently verify solutions with the 7B models we used for generation and finetuning. [3]Again, whilst this may be possible with more work, we found it very difficult to have models consistently grade CoTs that yielded answers close to, but not exactly, the right answer.

list and derivation of the reward functions we investigate can be found in Appendix A. Specifically, we investigate the three baseline values:

1. Average reward: $\frac{1}{n}\sum_{i=1}^{n} R(\mathbf{p}, \mathbf{r_i}, \mathbf{s})$, where $\mathbf{r_i}$ are multiple generated CoTs.

2. Empty CoT reward: $R(\mathbf{p}, \text{" "}, \mathbf{s})$, where the CoT is an empty string.

3. Random CoT reward: $R(\mathbf{p}, \mathbf{r_{random}}, \mathbf{s})$, where $\mathbf{r_{random}}$ is a sequence of random tokens.

In the main text of this paper, we focus on two main combinations of these augmentations (Appendix B.1 contains results for additional reward functions):

- **Delta Loss:** $R_{\text{DL}} = R(\mathbf{p}, \mathbf{r}, \mathbf{s}) - R(\mathbf{p}, \text{" "}, \mathbf{s})$, where we subtract the "Empty CoT" baseline.

- **Reasoning Advantage (RA):** $R_{\text{RA}} = \frac{R_{\text{clipped}}(\mathbf{p}, \mathbf{r}, \mathbf{s}) - R_{\text{clipped}}(\mathbf{p}, \text{" "}, \mathbf{s})}{R_{\text{clipped}}(\mathbf{p}, \text{" "}, \mathbf{s})}$, which is clipped delta loss normalized by the "Empty CoT" baseline.

We find that **Reasoning Advantage (RA)** is particularly effective. It satisfies each of the identified criteria in Table 1 and, in Section 5.1, we empirically demonstrate that RA can best distinguish effective CoT and pick out useful locations for CoT reasoning. Moreover, in Section 5.2, we demonstrate that RA is the only reward function which enables self-improveming CoT reasoning on free-form QA data, a key step towards self-improving CoT at on unstructured, pretraining-scale text.

## 5 EXPERIMENTS

### 5.1 REWARD FUNCTIONS FOR SELECTING WHAT & WHERE TO REASON

In this section, we empirically investigate a fundamental problem when self-improving CoT reasoning on unstructured, pretraining text: **What constitutes a suitable reward function for reasoning during general language model pretraining?** Building on the reward function criteria Section 4, we empirically investigate how different reward functions affect *what* and *where* reasoning is rewarded. Our two experiments reveal critical shortcomings in commonly used reward functions and demonstrate the advantages of our novel Reasoning Advantage (RA) function in addressing these limitations.

*What* **reasoning is rewarded**   This first experiment evaluates the ability of different reward functions to distinguish between three categories of CoTs: correct, incorrect, and randomly generated. We select 1,000 prefix-suffix pairs from random locations in the FineWeb text corpus of unstructured web-text data (Penedo et al., 2024). Then, for each pair, we generate the three types of CoT: corrent, incorrect, and random. "Correct" CoTs are generated using GPT-4o with post-rationalization—by showing GPT-4o both the prefix and suffix, but instructing the model to generate a CoT without explicitly repeating the suffix (similar to Zelikman et al. (2022)). "Incorrect" CoTs are generated by GPT-4o without post-rationalization—while these CoTs often exhibit sophisticated reasoning, they typically do not predict the exact suffix as well as the "correct" CoTs, which is enough for the purposes of this experiment. Finally, "random" CoTs consist of strings of random words and serve as our baseline. The goal is to evaluate how well different reward functions can rank these CoT types, with the ideal ordering: correct > incorrect > random.

To evaluate how well a reward function distinguishes between these CoT types, we compute the reward score for all CoTs—using Mistral-7B-Instruct (Jiang et al., 2023) to compute the log probabilities—and partition them into thirds: classifying the top third as "correct," the middle third as "incorrect," and the bottom third as "random." An effective reward function should rank the CoTs in the ideal order: correct > incorrect > random. The results in Table 2 demonstrate that RA performs best among loss, delta loss, and RA. Moreover, Table 3 shows results for the complete list of evaluated reward functions. Notice that while RA without normalization performs just slightly better, the normalized version significantly outperforms all other functions in the *where* experiments below. Hence, we pick the normalized version as our proposed reward function. Table 3 also shows the "Average reward" baseline, which is used Quiet-STaR (Zelikman et al., 2024), performs poorly in this setting—due to a lack of variation in reward over different CoTs.

| Reward Function | *What* (Acc) | *Where* (AUC) |
|---|---|---|
| RA | **66.3** | **77.0** |
| Delta Loss | 58.3 | 64.4 |
| Loss | 44.6 | 39.4 |

Table 2: Reward function performance for distinguishing CoT types (*What*) and identifying optimal CoT placement (*Where*). See Appendix B.1 for full results and confidence bounds.

The histogram in Figure 1 reveals that standard loss struggles primarily in distinguishing between "incorrect" and "random" CoTs. Interestingly, when we simplify to binary classification between only "correct" and "incorrect" CoTs, non-clipping methods perform similar to clipping methods, which suggests that the main advantage of clipping lies in distinguishing truly random reasoning.

***Where* reasoning is rewarded**   Next, we investigate how different function reward reasoning at different locations in a document. Using 1,000 problems each from GSM8K (Cobbe et al., 2021), CSQA (Talmor et al., 2018), and MMLU (Hendrycks et al., 2020), we first format each problem's question, multiple choice options, and answer as a single text string. We then create four (prefix, suffix) pairs per problem by splitting at different points: 1) mid-question, 2) after the question but before the multiple choice options, 3) after the multiple choice options (*the ideal location for CoT reasoning*), and 4) mid-answer. This setup aims to mimic a key fact about unstructured pretraining text: not all locations are suitable for CoT reasoning. That is, reasoning may be unhelpful if produced too early (insufficient context) or too late in a document.

To evaluate each reward function, we frame this as a binary classification task: identifying the ideal location (after the multiple choice answers but before the solution) versus the three suboptimal locations. Using reward as a classifier and computing the AUC for this classification task. We find that RA performs best, followed by delta loss and standard loss (see Table 2). Notice that functions which use a baseline consistently outperform those without, with clipping providing additional improvement. Particularly, subtracting the "Empty CoT" baseline helps distinguish between locations that have low loss due to effective CoT reasoning versus locations that have low loss because the suffix is trivial to predict without any reasoning (i.e., with an empty CoT). This partially explains why standard loss performs so poorly: it favors locations halfway through the answer where suffix prediction becomes trivial. Table 4 shows results for the complete list of evaluated reward functions.

**Summary**   Across both experiments, the Reasoning Advantage (RA) reward function outperformed standard loss and delta loss. As for the two main augmentations, clipping and baseline, we can summarize their effects. Clipping is often beneficial, almost never harmful, and requires minimal extra computation. We explore the impact of different clipping values in Appendix B.1 (Figure 5). And incorporating a baseline value provides a substantial boost in performance—especially the "Empty CoT" baseline. Moreover, a key advantage of the "Empty CoT" baseline is that it doesn't require generating any additional CoTs per (prefix, suffix) pair. In contrast, the "Average CoT" baseline requires taking the average loss over multiple CoTs for a single location. Appendix B.1 contains tables which show results for all combinations of augmentations. Notice that two combinations and the non-normalized version of RA performed slightly better on the *what* experiments, but they performed much worse on the *where* experiments. RA is the only function with strong performance on both tasks.

## 5.2   LEARNING TO REASON ON FREE-FORM QA DATA

To investigate the ability of different reward functions to facilitate self-improving CoT during pretraining, we create a new "free-form" QA dataset called MMLU-FREE-FORM by adapting the pop-

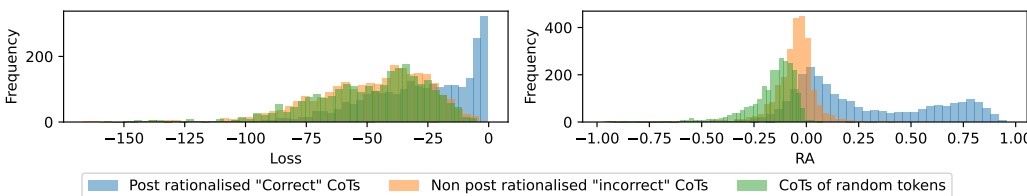

Figure 1: (*What* to reward) Distribution of reward scores across different CoT types using standard loss (left) and RA (right). Each histogram shows reward distribution: "correct" post-rationalized CoTs (blue), "incorrect" non-post-rationalized CoTs (orange), and "random" token CoTs (green). Notice, RA can better differentiate between incorrect and random CoTs. See details in Section 5.1.

ular MMLU training dataset (Hendrycks et al., 2020) to be closer to the unstructured text setting. Specifically, by removing its multiple-choice format and requiring models to generate full, unstructured answers—which are hard to verify. We use the entire labeled free-form solution as the suffix when computing rewards. This induces many of the challenges found in reasoning on unstructured text. For one, problems often become significantly more difficult to answer without multiple choice options, mirroring the complexity of next-token prediction in pretraining text. In some cases, the problems become almost impossible to answer (e.g., "Which of the following is the correct method to multiply 32 x 18?"). Moreover, the free-form nature of answers introduces substantial variance in response length and structure, making it challenging to predict an answer exactly. Finally, the same correct answer can be expressed in numerous valid ways (e.g., "Henry VIII had 6 wives" versus "In total there were 6 different women who were married to Henry the Eighth"). Without a list of multiple-choice options, it is unclear which answer should preferred. These challenges make the MMLU-FREE-FORM more representative of real-world pretraining text corpora.

Our purpose in creating MMLU-FREE-FORM is to make the smallest possible change to MMLU that reveals the limitations of existing reward functions. It acts as an intermediate benchmark between improving CoT reasoning using curated (question, CoT, answer) datasets and the challenging, unsolved task of self-improving CoT reasoning on unstructured text. Moreover, this dataset provides a higher density of clear opportunities for CoT reasoning compared to typical pretraining corpora, since we know that reasoning is particularly beneficial when predicting answers to questions, and prior works have shown that LLM reasoning ability on MMLU can be improved with only few thousand labeled CoT examples. Thus, for the purposes of our investigations, MMLU-FREE-FORM enables a more compute and time efficient study of reward functions, acting as a stepping stone towards self-improving CoT reasoning on the type of truly unstructured text seen during pretraining (i.e., OpenWebMath (Paster et al., 2023)).

We will release MMLU-FREE-FORM to the research community, and we hope it will serve as a helpful intermediate benchmark for future work to progress toward the unsolved problem of self-improving CoT reasoning on unstructured, pretraining-scale text. Further discussion about MMLU-FREE-FORM can be found in Appendix B.2.

Now, to self-improve CoT reasoning using MMLU-FREE-FORM as our dataset, we utilize a simple offline RL method. First, we generate 16 CoTs for each question (using Mistral-7B-Instruct with a temperature value of 0.5) and compute the reward for each CoT using the entire labeled free-form solution as the suffix. Then, we filter the CoTs with the highest reward (Dong et al., 2023), finetune on MMLU-FREE-FORM containing these self-inserted CoTs, and evaluate the trained model on a held-out test set. Notice that since all self-inserted CoTs are the same for each reward function, we can directly and efficiently compare each of them.

We test this pipeline using Mistral-7B (Jiang et al., 2023) and find that **only RA facilitates generalization**—both on the in-domain MMLU test set (see Figure 2a) and on zero-shot transfer to GSM8k (Cobbe et al., 2021) (see Figure 2b). These figures show the probability of the answer given the question and the generated CoT. This metric is also known as "expected accuracy", since it estimates how often the model would generate the exact ground truth answer if we repeatedly sampled completions given the question and CoT reasoning. We produce 95% confidence intervals through bootstrapping (LaFlair et al., 2015).

In more detail, only RA is able to substantially increase the answer probability on the MMLU test set, while filtering CoTs by standard loss, delta loss, or just randomly, improves test performance

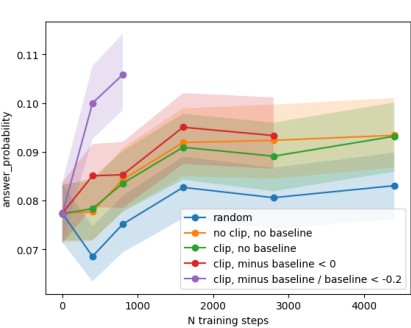

(a) MMLU Test Performance

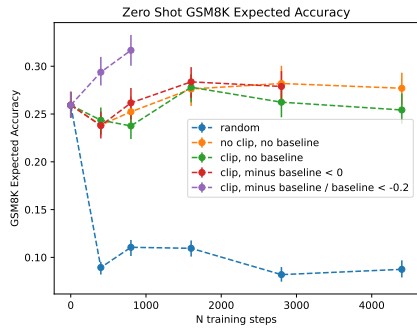

(b) GSM8K Transfer Performance

Figure 2: Reward function performance for self-improving reasoning on MMLU-FREE-FORM. Only RA (purple) facilitates generalization to MMLU test set and zero-shot transfer to GSM8k. Functions yield different amounts of filtered data (so different "N training steps"). '<' shows the filtering threshold, all baselines are "Empty CoT", and "random" means randomly picking CoTs.

by just a few percent and plateaus quickly with more steps. A full breakdown of in-domain MMLU performance is shown in Figure 7 in Appendix B.2. Moreover, only RA facilitates zero-shot transfer to GSM8K math problems—improving accuracy on by nearly 7%, compared to barely 0.5% when trained with other reward functions. Notice that we were only able to train for 1,000 steps with RA, since only 1,000 steps worth of generated CoTs were above the threshold of 0.2.

These strong results demonstrate that the resulting model learns generalizable reasoning—beyond just matching specific token patterns in the data. Thus, by rewarding CoTs that best reduce some form of loss on a suffix, we can enhance a model's general reasoning ability. This aligns with recent work (Du et al., 2024) showing that optimizing for loss during general pretraining improves downstream reasoning performance. Moreover, this shows that RA's key modifications to standard loss (clipping, adding a baseline, and normalizing) are crucial for learning generalizable reasoning.

## 6 CHALLENGES AND FUTURE DIRECTIONS

As it becomes increasingly challenging and expensive to curate large-scale (question, CoT, answer) datasets (Rein et al., 2023), the reasoning community has begun focusing on the challenging task of self-improving CoT reasoning on unstructured, pretraining-scale text. Our work frames this challenging task as an RL problem and demonstrates the effectiveness of RA at identifying useful reasoning, determining useful locations for reasoning, and facilitating self-improvement in the simplified MMLU-FREE-FORM setting. There is still more work to be done in order to solve the full, unstructured pretraining setting. In this section, we present an exploratory experiment that provides key insights into the barriers that must be overcome to achieve self-improvement at pretraining scale.

Specifically, we attempt to use our novel Reasoning Advantage (RA) reward function with the offline RL procedure from Section 5.2 to self-improve CoT reasoning on OpenWebMath (Paster et al., 2023), a pretraining corpus of unstructured web-text data. The two main steps of this procedure are: (1) generate a large batch of CoTs and self-insert them into OpenWebMath, and (2) finetune on the CoTs with the highest reward scores. Our analysis indicates that this method is not sufficiently powerful to escape the local optimum of extremely conservative CoT reasoning that just summarizes previous information instead of attempting to actually solve problems (see Appendix C for examples). In Section 6.1, we provide key insights into why this method is not sufficient. In Section 6.2, we provide a more detailed experimental setup and additional results.

### 6.1 KEY INSIGHTS (NEW SUBSECTION)

To understand why this procedure fails on OpenWebMath, we isolate the problem into two key components: generating diverse CoTs and identifying useful CoTs. In our offline RL approach, the role of the reward function is purely to *identify* useful CoTs to use for training, and Section 5.1 demonstrates that RA excels at this task. This suggests that the remaining challenge lies in generation. Indeed, our analysis shows that only 0.01% of the generated CoTs achieve a reward above 0.2, which is our filtering threshold for RA (a decent threshold for "good reasoning" in our experience). Moreover, many of the CoTs that passed the filtering threshold exhibited the conservative strategy

described previously: they simply summarize past information from the context. This explains why the model learned to be overly conservative. However, these overly conservative CoTs which made it past the RA threshold were still superior to those that did not pass the threshold (those ones mainly contained **incorrect** reasoning that predicted the subsequent tokens incorrectly). This indicates that RA actually succeeded at its job of identifying the best reasoning from the generated batch of CoTs, and that the main issue indeed lies with the *lack of diversity* in the generated CoTs.

Thus, to facilitate self-improvement using RA, we must crucially generate a diverse-enough set of CoTs so that there are enough useful samples for RA to identify. This remains a critical barrier for future work to investigate. To increase the diversity of explored CoTs, future work might use Quality-Diversity (Mouret & Clune, 2015) or other evolutionary techniques (Fernando et al., 2023; Samvelyan et al., 2024), which could generate more diverse CoTs. It would also be worth exploring different prompting strategies (we used a single system prompt to generate these CoTs, and did not spend much time prompt engineering). Better exploration may also be facilitated by using online RL, but the only existing method in this direction generates a CoT at every token in a document (Zelikman et al., 2024), which is highly inefficient. Thus, we believe that our computationally feasible offline RL approach of generating CoTs in large, offline batches and performing supervised fine-tuning is key to enabling the self-improvement of CoT reasoning at the pretraining scale. However, future work should investigate ways of generating a more diverse batch of CoTs in order for this method to work. One possible idea is to use a combination of RHO-1 (Lin et al., 2024) and RA.

To encourage future research in this direction, we will open-source our offline RL code, which runs on an academic compute budget. We will also open-source MMLU-FREE-FORM, which we believe acts as a useful intermediate benchmark between curated QA data and general language pretraining.

### 6.2 EXPERIMENTAL SETUP AND ADDITIONAL RESULTS (NEW SUB SECTION)

We first finetune Mistral-7B-Instruct (Jiang et al., 2023) on a small set of CoTs to learn the "[THOUGHT]...[/THOUGHT]" syntax. Then, we randomly sample 50,000 (prefix, suffix) pairs from OpenWebMath and generate CoTs for each location using Mistral-7B-Instruct with a temperature value of 0.5. From this pool of generated CoTs, we create three variants of an augmented OpenWebMath dataset by selecting 3,200 CoTs using different filtering methods: (1) random selection, (2) best loss scores, and (3) best RA scores.

Throughout training, we evaluate each model's CoT reasoning ability on a holdout set of OpenWebMath documents. At each checkpoint, we identify locations where "[THOUGHT]" is the predicted next token, generate CoTs at these points, and measure three metrics on the holdout documents (excluding CoT tokens but using them as context): standard loss, delta loss, and RA. Figure 8 show three plots—each measuring one of these metrics at various checkpoints throughout training. Notice that each line represents an entirely different model trained on differently filtered CoTs.

## 7 CONCLUSION (UPDATED, BETTER CLARITY OF CONTRIBUTIONS)

As it becomes increasingly challenging and prohibitively expensive to curate large-scale (question, CoT, answer) datasets, the LLM reasoning community has began to focus on the challenging task of self-improving CoT reasoning on unstructured text at the pretraining scale. We frame this as a reinforcement learning problem and investigate a fundamental question: What constitutes a suitable reward function for learning to reason during general language model pretraining? We outline the desirable qualities of such a reward, point out critical shortcomings in many commonly used reward functions, and introduce Reasoning Advantage (RA), a novel reward function which addresses these limitations. Further, we provide a comprehensive analysis on how different functions affect: (1) the ability to identify effective CoT reasoning (*what* reasoning is rewarded), and (2) the ability to pick out useful locations to produce CoT reasoning (*where* reasoning is rewarded). To our knowledge, our work is the first to provide this type of analysis on reward functions for self-improving CoT reasoning on unstructured text.

We introduce MMLU-FREE-FORM, a small step towards the full unstructured pretraining setting, and demonstrate that only RA is able to facilitate generalization when self-improving CoT reasoning on MMLU-FREE-FORM. There is still more work to be done in order to solve the full, unstructured pretraining setting, and we present an exploratory experiment that provides key insights into the barriers that must be overcome to achieve self-improvement at pretraining scale. Most importantly, future work should investigate methods for generating a more diverse set of CoTs. We will open source all of our code, which runs on academic compute, to facilitate future work in this direction.

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

## A  FORMAL DEFINITIONS

We look at the following metrics for evaluation of intermediate contemplation for next token prediction, that is, given a prefix set of tokens $p = p_1, ..., p_n$, a generated set of intermediate reasoning tokens $r$ and a suffix of $m$ tokens to predict $s = s_1, ...., s_m$, produce a score $R(p, r, s) \in \mathbb{R}$. We use $P(s_0|p + r)$ to denote the probability distribution over all tokens on the first token of the suffix.

1. **Accuracy (using generation)**: Generate, such as through sampling or via greedy decoding, $k$ continuations $\hat{s_1}, ..., \hat{s_k}$ of length $n_s$ from the input $p + r$.

$$R_{\text{accuracy using generation}} = \frac{1}{k} \sum_{i=1}^{k} \mathbb{I}[\hat{s}_i = s] \tag{4}$$

2. **Accuracy (generation free)**: Accuracy using generation requires at least $n_s$ forward passes. Instead, one can leverage the autoregressive nature of transformers to obtain the probability distribution over next tokens for the entire answer simultaneously. That is input the model $p + r + s$ and obtain $P(\hat{s_0}|p + r), P(\hat{s_1}|p + r + s_0), ...$ with one forward pass. Looking at whether the argmax of this distribution is $s$ is equivalent to accuracy above using greedy decoding, and taking $P(s|p + r)$

$$R_{\text{expected greedy accuracy}} = \Pi_{i=1}^{n_s} \mathbb{I}[\arg\max(P(\hat{s}_i|p + r + s_{:i})) = s_i] \tag{5}$$

$$R_{\text{expected accuracy}} = \Pi_{i=1}^{n_s} P(\hat{s}_i|p + r + s_{:i}) \tag{6}$$

3. **Loss**: We use the cross entropy loss over tokens, i.e:

$$R_{\text{cross entropy loss}} = -\sum_{i=1}^{n_s} \log(P(\hat{s}_i|p + r + s_{:i})) \tag{7}$$

4. **Delta Loss:** The difference in cross entropy between using and not using the reasoning.

$$R_{\text{delta cross entropy loss}} = -\sum_{i=1}^{n_s} \log(P(\hat{s}_i|p + r + s_{:i})) - -\sum_{i=1}^{n_s} \log(P(\hat{s}_i|p + s_{:i})) \tag{8}$$

5. **Normalised Delta loss:** Different answers have varying levels of inherent predictability. Thus desirable values for loss or delta loss can vary massively. To account for this, we divide by the answer likelihood without reasoning.

$$R_{\text{normalised delta cross entropy loss}} = R_{\text{delta cross entropy loss}} / -\sum_{i=1}^{n_s} \log(P(\hat{s}_i|p + s_{:i})) \tag{9}$$

6. **Clipped variants**: We evaluate loss, delta loss and normalised delta loss with clipping applied to the token log probabilities to prevent large values dominating. Our final results leverage $\epsilon = -3$. For example

$$R_{\text{clipped loss}} = -\sum_{i=1}^{n_s} \max[\log(P(\hat{s}_i|p + r + s_{:i})), \epsilon] \tag{10}$$

7. Normalised clipped delta loss (**Reasoning Advantage**): We combine the benefits of delta loss, normalisation and clipping into one metric.

8. **LLM-as-judge**: Generate, such as through sampling or via greedy decoding, $k$ continuations $\hat{s_1}, ..., \hat{s_k}$ of length $n_s$ from the input $p + r$. Let $M(p, r, \hat{s}_i, s_i)$ denote whether a model considers $\hat{s}_i$ to match be the correct answer of $s_i$.. Average over the $k$ completions, i.e:

$$R_{\text{Model eval}} = \frac{1}{k} \sum_{i=1}^{k} M(p, r, \hat{s}_i, s_i) \tag{11}$$

## B ADDITIONAL EXPERIMENT DETAILS, RESULTS, AND VISUALIZATIONS

To compute the log probabilities for all reward functions, we used Mistral-7B-Instruct (Jiang et al., 2023) finetuned on a small set of 1,000 GPT-4 generated CoTs that have been filtered for correctness (by providing the model with the correct answer and asking whether it corresponds). This finetuning allows us to start from a base model that is used to the format of:

```
### Question: <question> ### Thought <reasoning> ### Answer: <response>.
```

### B.0.1 ADDITIONAL VISUALIZATIONS FOR WHAT & WHERE TO REASON

Figure 3 and Figure 4 provide additional visualization for the *What* and *Where* experimental results from Section 5.1, respectively.

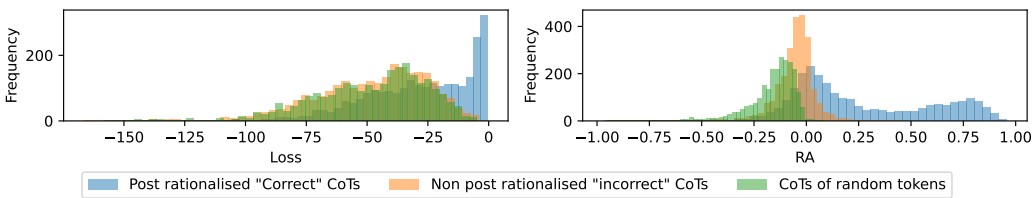

Figure 3: (*What* to reward) Distribution of reward scores across different CoT types using standard loss (left) and RA (right) reward functions. Each histogram shows the reward distribution for three categories: "correct" post-rationalized CoTs (blue), "incorrect" non-post-rationalized CoTs (orange), and "random" token CoTs (green). Notice that RA is better able to differentiate between incorrect and random CoTs. Moreover, the RA scores are normalized to the range [-1, 1], which may facilitate better learning. See Section 5.1 for more details.

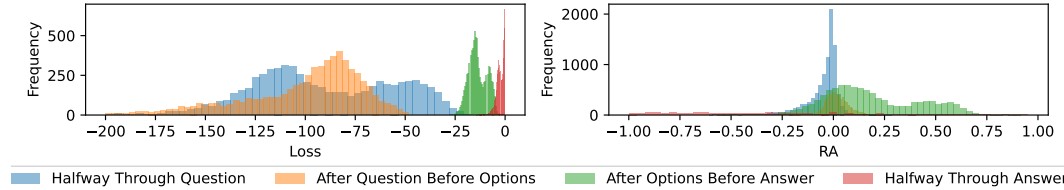

Figure 4: (*Where* to reward) Distribution of reward scores for CoTs inserted at different locations using standard loss (left) and RA (right) reward functions. Each histogram shows the reward distribution for four insertion points: halfway through question (blue), after question before multiple-choice options (orange), after multiple-choice options before answer (green), and halfway through answer (red). As mentioned in Section 5.1, we assume that after multiple-choice options before answer (green) is the optimal location to generate CoT reasoning. RA successfully scores CoTs generated at this location higher, while standard loss does not. Particularly, standard loss fails to prevent halfway-through-answer CoTs from receiving high rewards.

### B.1 WHAT & WHERE TO REASON RESULTS FOR ADDITIONAL REWARD FUNCTIONS

Table 3 and Table 4 show full results for additional reward functions. That is, for the entire family of loss-based reward functions. Moreover, they include results for the "empty CoT" baseline as well as the "random CoT" and "mean loss" baselines. We explore incorporating these baselines both with normalization $(R - B)/B$ and without normalization $R - B$, where $R$ is the reward score and $B$ is the baseline value.

In Tables 3 and 4, RA outperforms standard loss and delta loss—as in the main text. However, it's worth mentioning that there are three combinations of augmentations that perform better than RA in Table 3 (*What* to reward), while performing much worse than RA in Table 4 (*Where* to reward). In the main text of this work, we choose to focus mainly on standard loss, delta loss, and RA since delta

loss shows how the simple change of adding an empty CoT baseline improves results over standard loss, and RA shows the added effectiveness of clipping and normalization.

| Name | Baseline | Clipping | Normalisation | Mean | q0.025 | q0.975 | Rank |
|---|---|---|---|---|---|---|---|
| Loss | none | none | none | 44.6% | 44.0% | 45.4% | 9 |
| - | empty CoT reward | clipped | none | 67.2% | 65.7% | 68.3% | 3 |
| **RA** | **empty CoT reward** | **clipped** | **yes** | **66.3%** | **64.5%** | **67.8%** | **4** |
| Delta Loss | empty CoT reward | none | none | 58.3% | 57.8% | 58.9% | 8 |
| - | empty CoT reward | none | yes | 58.8% | 58.1% | 59.8% | 7 |
| - | random CoT reward | clipped | none | 80.4% | 79.7% | 81.4% | 1 |
| - | random CoT reward | clipped | yes | 78.4% | 77.8% | 79.0% | 2 |
| - | random CoT reward | none | none | 60.9% | 60.1% | 62.7% | 6 |
| - | random CoT reward | none | yes | 60.9% | 59.2% | 63.1% | 5 |
| - | average reward | clipped | none | 30.8% | 30.1% | 31.7% | 10 |
| - | average reward | clipped | yes | 30.7% | 29.9% | 31.3% | 11 |
| - | average reward | none | none | 29.2% | 28.7% | 29.8% | 13 |
| - | average reward | none | yes | 30.7% | 30.0% | 31.7% | 11 |

Table 3: Full results for *What* to reward experiment, showing all combinations of augmentations to the basic loss-based reward in Equation 3.

| Name | Baseline | Clipping | Normalisation | Mean | q0.025 | q0.975 | Rank |
|---|---|---|---|---|---|---|---|
| Loss | none | none | none | 39.4% | 37.7% | 40.8% | 6 |
| - | empty CoT reward | clipped | none | 55.9% | 52.5% | 59.9% | 4 |
| **RA** | **empty CoT reward** | **clipped** | **yes** | **77.0%** | **75.3%** | **79.0%** | **1** |
| Delta Loss | empty CoT reward | none | none | 64.4% | 62.7% | 67.0% | 3 |
| - | empty CoT reward | none | yes | 73.0% | 71.9% | 74.3% | 2 |
| - | random CoT reward | clipped | none | 29.8% | 28.2% | 30.6% | 9 |
| - | random CoT reward | clipped | yes | 40.8% | 38.9% | 43.4% | 5 |
| - | random CoT reward | none | none | 27.9% | 26.7% | 28.8% | 11 |
| - | random CoT reward | none | yes | 27.3% | 25.8% | 28.6% | 13 |
| - | average reward | clipped | none | 27.7% | 25.8% | 29.2% | 12 |
| - | average reward | clipped | yes | 33.4% | 32.5% | 35.4% | 7 |
| - | average reward | none | none | 28.3% | 26.5% | 30.0% | 10 |
| - | average reward | none | yes | 32.1% | 30.8% | 33.4% | 8 |

Table 4: Full results for *Where* to reward experiment, showing all combinations of augmentations to the basic loss-based reward in Equation 3.

| | What to Contemplate (Accuracy) | | | Where to Contemplate (AUC) | | |
|---|---|---|---|---|---|---|
| Method | Mean | $q_{0.025}$ | $q_{0.975}$ | Mean | $q_{0.025}$ | $q_{0.975}$ |
| RA | **0.546** | **0.530** | **0.563** | **0.875** | **0.857** | **0.890** |
| Delta Loss | 0.498 | 0.484 | 0.511 | 0.684 | 0.668 | 0.700 |
| Loss | 0.439 | 0.426 | 0.453 | 0.386 | 0.367 | 0.401 |

Table 5: Reward function performance using Llama-3.1-8B (Dubey et al., 2024) (in contrast to Mistral-7B as in the main body) for distinguishing CoT types (*What*) and identifying optimal CoT placement (*Where*). This agrees with the results for Mistral-7B in the main body.

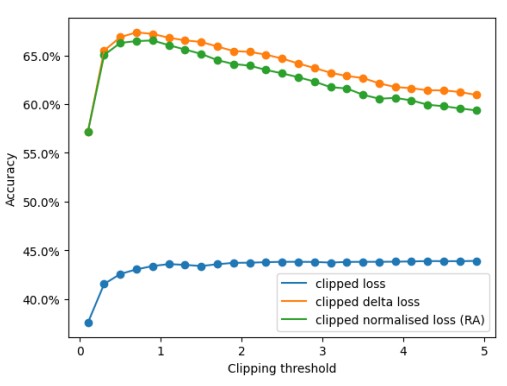
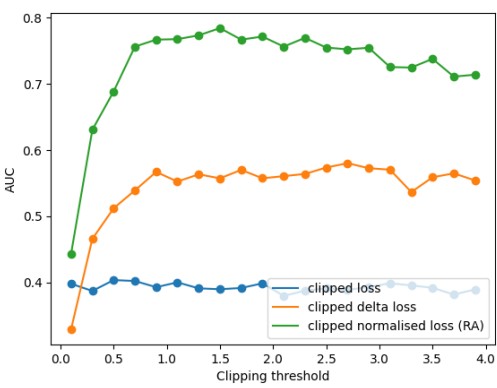

(a) Ablation of clipping value for distinguishing CoT types (What to reason) with Mistral-7B.

(b) Ablation of clipping value for identifying optimal CoT placement (Where to reason) with Mistral-7B.

Figure 5: Ablating the clipping value used in RA. A value of 1.0 is reasonably optimal for both experiments, and was therefore used for the results in Table 2 and the MMLU-Free-Form experiments.

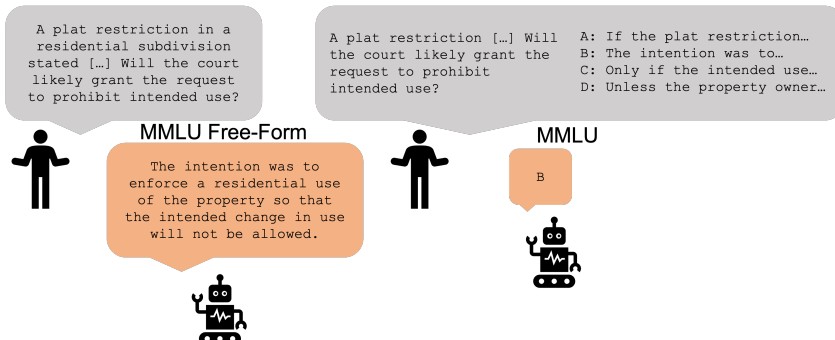

Figure 6: Example from MMLU-FREE-FORM, our modified version of MMLU (Hendrycks et al., 2020) designed to study improving CoT reasoning on unstructured, open-ended text. By removing multiple-choice options, answers become free-form so that they can can be expressed in multiple different and equally valid ways—this invalidates the use of accuracy without an external verifier. The left-hand side is the example from MMLU-FREE-FORM and the right-hand side is the original example from MMLU.

## B.2 PERFORMANCE BREAKDOWN FOR SELF-IMPROVING COT REASONING ON MMLU-FREE-FORM

Figure 7 shows a more complete breakdown of the results on the MMLU test set after self-improving CoT reasoning on MMLU-FREE-FORM using the method outlined in Section 5.2. The "reasoning style questions" require quantitative reasoning and span a wide range of subjects including physics, biology, accounting, mathematics, and computer science. Moreover, we observe far greater improvement on "reasoning style questions" compared to "recall style questions". This interesting result makes sense, since additional reasoning doesn't help as much when trying to recall a fact that was present in the context.

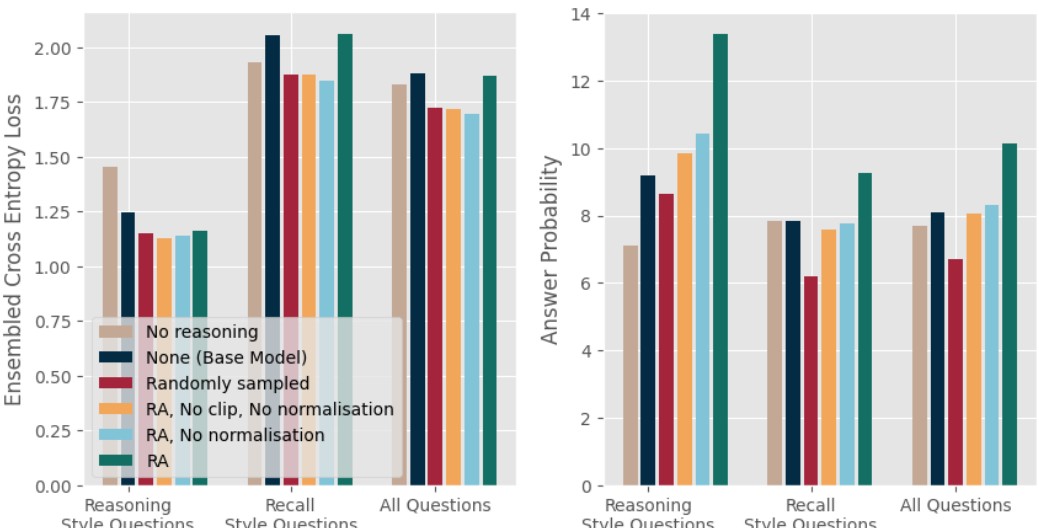

Figure 7: Performance breakdown on MMLU test set after self-improving CoT reasoning on MMLU-FREE-FORM. Results are shown for different question types. Left: Ensembled cross-entropy loss (higher is better), computed as average log-likelihood across multiple CoTs. Right: Answer probability (higher is better). See Section 5.2 for full experiment and method details.

## C    EXAMPLES OF CONSERVATIVE CHAIN-OF-THOUGHT

As discussed in Section 6, our offline RL procedure applied to the unstructured OpenWebMath dataset (Paster et al., 2023) converges to a local optimum where the model generates overly conservative CoT reasoning, merely restating or summarizing information rather than attempting problem-solving. Below are some examples of this behavior. Notice that the model knows about the "prefix" and "completion" from the prompts it received.

**GSM8K Example:**

```
Q: Richard lives in an apartment building with 15 floors. Each
   floor contains 8 units, and 3/4 of the building is occupied.
   What's the total number of unoccupied units In the building? [
   THOUGHT] The prefix provides information about an apartment
   building with 15 floors, each floor containing 8 units, and
   3/4 of the building occupied. The completion likely provides
   the total number of unoccupied units in the building, based on
    the given information. [/THOUGHT]
A: 30
```

**OpenWebMath Example 1:**

```
In this article, by using norms($T$ and $C$), we present the
   concept of intuitionistic fuzzy implicative ideals,
   intuitionistic fuzzy closed implicative ideals and
   intuitionistic fuzzy commutative ideals of $BCI$-algebras.
   Some interesting results of them are given. Characterisations
   of implicative ideals, closed implicative ideals and
   commutative ideals of $BCI$-algebras by using them are
   explored. By using intersections, direct products and
   homomorphisms, some interesting results are obtained [THOUGHT]
```

> Consider the prefix, which introduces the concept of intuitionistic fuzzy implicative ideals in $BCI$−algebras. The completion likely provides a definition or description of these ideals, along with their properties and applications. [/THOUGHT].

**OpenWebMath Example 2:**

# Chapter 8 − Polynomials and Factoring − Chapter Review − 8−3 and 8−4 Multiplying Binomials: 31

$9r^{2}−12r+4 [THOUGHT] The prefix provides a list of equations, each with a variable and a coefficient. The completion likely provides the solution to each equation, using the variable and coefficient to determine the value of the equation. The completion may also provide a step−by−step explanation [/THOUGHT] $

#### Work Step by Step

Simplify and write in standard form $(3r−2)^{2}$ Rewrite as: $(3r−2)(3r−2)$ Foil $9r^{2}−6r−6r+4$ Combine like terms $9r^{2}−12r+4$

After you claim an answer you'll have 24 hours to send in a draft. An editor will review the submission and either publish your submission or provide feedback.

## D  SOCIETAL IMPACT

While our work is primarily analytical and does not introduce new models, the broader direction of self-improving CoT reasoning on large-scale unstructured text datasets could significantly enhance LLMs' problem-solving capabilities—if successful. Such advances would amplify both the benefits and risks associated with current language models, warranting continued attention from the research community on ensuring responsible development.

## E  ADDITIONAL VISUALIZATIONS

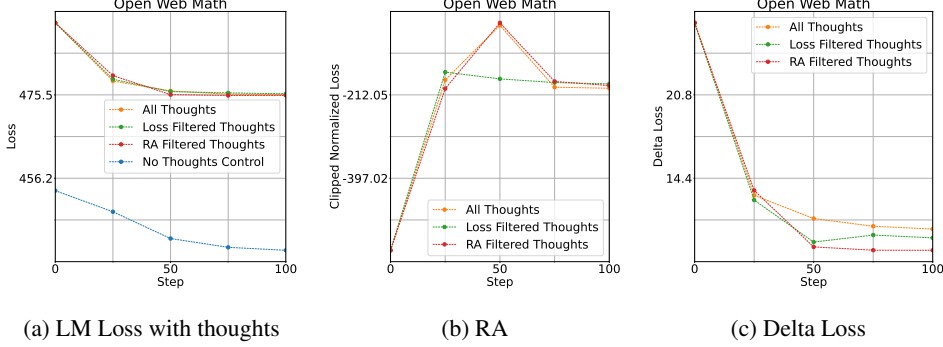

(a) LM Loss with thoughts      (b) RA      (c) Delta Loss

Figure 8: Standard loss, delta loss, and RA on the *holdout* documents measured at different training checkpoints (see Section 6.2 for details). Each line represents an entirely different model trained on differently filtered CoTs. The filtering strategies are: random selection ("All Thoughts"), loss-based ("Loss Filtered Thoughts"), RA-based ("RA Filtered Thoughts"), and a "No Thoughts Control" baseline (trained on standard OpenWebMath documents without any self-inserted CoTs).

