# OpenReview forum: "Towards Learning to Reason at Pre-Training Scale"
_ICLR.cc/2025/Conference — Submitted to ICLR 2025_

### Official Review · Reviewer_MbaU · 2024-10-20

**Soundness:** 3
**Presentation:** 3
**Contribution:** 3
**Rating:** 6
**Confidence:** 2

**Summary:**

The paper explores effective rewards that could be applied during LLM pretraining. Especially, the paper explores various reward functions based on what reasoning is learnt and where reasoning is rewarded. Based on the findings, the paper suggests, RA (Reasoning advantage) which facilitates self-improving CoT reasoning on free- form question-answering (QA) data.

**Strengths:**

- The paper provides useful insights for designing rewards for language model training.
- The authors explores the effectiveness of RA on multiple experimental settings.

**Weaknesses:**

- Unlike the motivation of the paper, the proposed method, RA, is not effective for pre-training scale, questioning the scalability of the proposed method.
- The paper measures the performance by using 'expected accuracy' metric, which makes comparison with other methods difficult. What is the absolute accuracy performance for Figure 4?
- The paper only uses a single backbone model to show the effect of the proposed method.

**Questions:**

- How effective is RA compared to another baseline model which is directly trained to predict the final answer without training to generate CoT?
- How much additional overhead occurs for applying RA during pre-training (Section 6)?

---

> ### Author Response · Authors · 2024-11-19
> **Response to Reviewer MbaU [1/2]**
>
> Thank you for your review and comments. We’re glad that the reviewer finds our work insightful for designing rewards for language model training.  Please see below for responses to your comments and questions.
>
> >Unlike the motivation of the paper, the proposed method, RA, is not effective for pre-training scale, questioning the scalability of the proposed method.
>
> Thanks for raising this point, we agree with the reviewer that our work does not solve the full unstructured, pretraining setting. We think it may be helpful to provide additional background, which we hope clarifies the contributions of our work:
>
> As it becomes increasingly challenging and prohibitively expensive to curate large-scale (question, CoT, answer) datasets, the LLM reasoning community has begun focusing on a grand challenge: self-improving CoT reasoning on unstructured text at pretraining scale.
>
> In this work, we introduce MMLU-FREE-FORM as a small step towards the grand challenge of truly unstructured text. As mentioned by Reviewer KyrV, it acts as “*an intermediate benchmark between structured QA and general language modeling*”. However, we show that even taking this small step renders current reward functions unusable. That is, standard reward functions cannot self-improve CoT reasoning *even in* this simplified free-form QA setting. **Our work introduces RA to address this problem**—it’s the only reward function which facilitates generalization when self-improving on MMLU-FREE-FORM. We also perform a comprehensive analysis of reward functions and how they affect *what* and *where* reasoning is rewarded. To our knowledge, our work is the first to provide this type of analysis on reward functions for self-improving CoT reasoning on unstructured text.
>
> To be clear, our work does not solve the grand challenge mentioned. Our work demonstrates that current reward functions fail for even the small step of MMLU-FREE-FORM, introduces a novel reward function that mitigates this problem, and performs a comprehensive analysis with insights to help facilitate future research in this direction. We believe these to be major contributions to the literature, and have dramatically updated our Introduction, Section 5, Section 6, and Conclusion with more targeted descriptions of our contributions. The key points are updated with red text.
>
> >The paper measures the performance by using 'expected accuracy' metric, which makes comparison with other methods difficult. What is the absolute accuracy performance for Figure 4?
>
> Thanks for the question. Expected accuracy is commonly used for evaluating language models on QA benchmarks, with many works simply referring to it as accuracy. For example, one of the most important recent works on self-improving LLM reasoning [1] reports accuracy, but it's only clear when going through their code that this is in fact expected accuracy.
>
> Given a question and a reasoning trace, we compute expected accuracy as the probability of the correct answer. Another way to compute accuracy is by greedy decoding a single answer, or by sampling many answers and averaging the performance, which approaches expected accuracy given enough samples. Usually, expected accuracy is preferred because it represents the raw distribution learnt by the model, without depending on a specific decoding procedure. We have added additional clarification regarding this point near the end of Section 5.2 (blue text).
>
> [1] Zelikman et al, Quiet-STaR: Language Models Can Teach Themselves to Think Before Speaking, 2024
>
> >The paper only uses a single backbone model to show the effect of the proposed method.
>
> We thank the reviewer for raising this point, and we agree that running experiments with another model would strengthen the paper. Therefore, we have repeated the *what* and *where* experiments from Section 5.1 with another model, LLama 3.1 8B. We present the results in Appendix B Table 5 and find them to be consistent with our original results using Mistral 7B in Table 2.

---

> ### Author Response · Authors · 2024-11-19
> **Response to Reviewer MbaU [2/2]**
>
> >How effective is RA compared to another baseline model which is directly trained to predict the final answer without training to generate CoT?
>
> Thank you for pointing out that this important baseline was missing. We have included an updated Figure 7 in the latest version of the manuscript which includes a “no reasoning” baseline of training to predict the answer without any CoT. This new baseline performs worse than RA and many of the other baselines (Figure 7, right-side). In particular, the difference in performance is largest for reasoning style questions.
>
> >How much additional overhead occurs for applying RA during pre-training (Section 6)?
>
> Our offline RL method has three main steps: (1) generate a large batch of CoTs, (2) self-insert them into the pretraining dataset, and (3) filter the ones with the highest scoring rewards and finetune the model. Thus, the main computational overhead occurs **before** training: generating the CoTs and using RA to score them. This can be efficiently parallelized, and thankfully, getting the score from RA only requires 2 forward passes (one with the CoT, and one for the “Empty CoT” baseline). During supervised finetuning, we just fit the dataset with the self-inserted CoTs as normal, which adds minimal overhead. So, the additional overhead is mostly **before** training, it can be massively parallelized (RA is parallelizable since it is loss-based), and it is relatively insignificant compared to the unavoidable cost of actually generating the CoTs.

---

> > ### Comment · Reviewer_MbaU · 2024-11-24
> >
> > Thank you for your clarification and additional experiments.
> > I will keep my score accordingly.

---

> > > ### Author Response · Authors · 2024-11-25
> > >
> > > Thank you again for providing initial comments that have helped to improve the paper. In light of our clarifications and additional experiments, we would appreciate it if you considered raising the confidence level in your score. And if you have any outstanding sources of uncertainty, please don’t hesitate to share these with us so that we can further improve the clarity of our work.

---

### Official Review · Reviewer_ZH4y · 2024-10-31

**Soundness:** 3
**Presentation:** 3
**Contribution:** 2
**Rating:** 5
**Confidence:** 4

**Summary:**

This paper explores a method for self-improving CoT reasoning in LLMs without relying on curated datasets. By leveraging reinforcement learning on general pre-training data, the authors aim to enhance models’ reasoning abilities across diverse tasks. They introduce a new reward function, Reasoning Advantage (RA), which better identifies effective reasoning, and demonstrate its impact on open-ended question-answering tasks. The paper highlights RA’s potential but also suggests that more advanced optimization methods are needed for scalable CoT improvements in broader, less structured contexts.

**Strengths:**

1. This paper addresses an important issue: achieving self-improvement of CoT reasoning during the pre-training phase. This approach has significant potential to help overcome the data bottleneck in LLMs.

2. The paper explores several types of reward functions and establishes criteria for an effective reward function, which is valuable and insightful for future research in this area.

**Weaknesses:**

1. The technical contributions of the paper are relatively weak. The proposed MMLU-FREE-FORM is merely a simple adaptation of the original MMLU, and the introduced RA is only a minor modification based on token loss.

2. The paper somewhat overstates its contributions. The authors primarily demonstrate the positive impact of RA on MMLU-FREE-FORM, yet MMLU-FREE-FORM is derived from the structured MMLU dataset and cannot be regarded as a typical pre-training dataset. In fact, experiments on OpenWebMath show minimal improvement. Typical pre-training datasets often include substantial noise, such as HTML elements, which is a key challenge in achieving self-improvement CoT during the pre-training phase.

3. The paper lacks discussion on relevant work in reasoning enhancement during the pre-training phase, such as https://arxiv.org/pdf/2404.07965.

4. The experiments are insufficiently comprehensive, as they are conducted on only one model and one dataset. Testing with models of different parameter sizes within the same series or different architectures could help demonstrate the generalizability of RA.

5. The presentation of the paper could be improved. Some key findings should be in the main body rather than the Appendix, such as Appendix D and the definition of RA in Appendix A. Essential parameters, like the type of LLM used and inference hyperparameters, should also be included in the main text.

    Minor:
    - Punctuation should be added at the end of each equation.
    - Some quotation marks are unmatched, such as in line 265 and line 349.
    - Figure 1 appears somewhat rudimentary.

**Questions:**

See above.

---

> ### Author Response · Authors · 2024-11-19
> **Response to Reviewer ZH4y [1/2]**
>
> We thank the reviewer for engaging deeply with the work. We’re glad that the reviewer found our investigations into effective reward functions for CoT reasoning both valuable and insightful.
>
> We have updated the paper to clarify our contributions and address the reviewer’s concerns, with the main updates in red text. We address specific questions and comments below.
>
> >The technical contributions of the paper are relatively weak. The proposed MMLU-FREE-FORM is merely a simple adaptation of the original MMLU, and the introduced RA is only a minor modification based on token loss.
>
> We agree that MMLU-FREE-FORM is not a substantial technical change from MMLU. However, our purpose for creating MMLU-FREE-FORM was not to create a radically new dataset, but to make the smallest possible change to MMLU that reveals the limitations of existing reward functions. It acts as an important middle-ground between improving CoT reasoning using curated (question, CoT, answer) datasets and the challenging, unsolved task of self-improving CoT reasoning on unstructured text. This is because MMLU-FREE-FORM does not allow for using exact-match accuracy as a reward metric (similar to unstructured pretraining text) and yet offers a higher density of clear opportunities for CoT reasoning compared to typical pre-training corpora, making it an ideal stepping-stone towards the ultimate goal of self-improving CoT reasoning on unstructured text. As mentioned by Reviewer KyrV, "*the creation of MMLU-FREE-FORM as an intermediate benchmark between structured QA and general language modeling is clever and useful for the research community. The empirical results showing successful transfer learning to GSM8K math problems provide concrete validation of their approach.*" We have updated the Introduction and Section 5.2 to clarify the contribution of MMLU-FREE-FORM, with the main updates in red text.
>
> We also agree with the reviewer that RA is a modification of token loss: by using clipping, subtracting the "empty CoT" baseline, and normalizing. As above, our goal was to make the smallest, simplest modification to the existing paradigm (standard loss) that has the potential to work for this setting. We believe that the contribution of RA is quite significant. It performs substantially better than token loss on the *what* to reward experiments (distinguishing effective CoT) and the *where* to reward experiments (picking out useful locations for producing CoT). Moreover, and possibly our most important result, **only RA is able to facilitate generalization to the MMLU test set and zero-shot transfer to GSM8K** when self-improving CoT reasoning on MMLU-FREE-FORM. In addition, we strongly believe that RA being based on token loss is a key advantage of this function. It requires only two forward passes, does not require an external strong model, is not limited to exact-match heuristics like accuracy-based functions (which fail on unstructured text), and allows the model to place weight over a distribution of valid answers. We have updated Section 4 to clarify this point, with the main updates in red text.
>
> We hope that our explanations helped to clarify the contributions of MMLU-FREE-FORM and the RA reward function.
>
> >The paper somewhat overstates its contributions. The authors primarily demonstrate the positive impact of RA on MMLU-FREE-FORM, yet MMLU-FREE-FORM is derived from the structured MMLU dataset and cannot be regarded as a typical pre-training dataset. In fact, experiments on OpenWebMath show minimal improvement.
>
> We thank the reviewer for their feedback on this point. Here, we aim to provide additional background, which we hope clarifies the important contributions of our work.
>
> Curating large, challenging, and diverse (question, CoT, answer) datasets for improving LLM reasoning has become exceptionally expensive (millions of dollars) and very challenging (requiring thousands of expert hours). With this as motivation, the LLM reasoning community has recently begun focusing on a grand challenge: self-improving CoT reasoning on unstructured text at pretraining scale.
>
> We agree with the reviewer that MMLU-FREE-FORM is not a typical pre-training dataset. It is a small step towards the grand challenge of truly unstructured text. However, we show that even taking this small step renders current reward functions unusable. That is, standard reward functions cannot self-improve CoT reasoning *even in* this simplified free-form QA setting. **Our work introduces RA to address this problem**—it’s the only reward function which facilitates generalization when self-improving on MMLU-FREE-FORM. We also perform a comprehensive analysis of reward functions and how they affect what and where reasoning is rewarded. To our knowledge, our work is the first to provide this type of analysis on reward functions for self-improving CoT reasoning on unstructured text.
>
> **[Continued in 2nd Response]**

---

> > ### Author Response · Authors · 2024-11-19
> > **Response to Reviewer ZH4y [2/2]**
> >
> > To be clear, our work does not solve the grand challenge mentioned. Our work demonstrates that current reward functions fail for even the small step of MMLU-FREE-FORM, introduces a novel reward function that mitigates this problem, and performs a comprehensive analysis to help facilitate future research in this direction. We believe these to be major contributions to the literature, and have dramatically updated our Introduction, Section 5, Section 6, and Conclusion with more targeted descriptions of our contributions. The key points are updated with red text.
> >
> > >The paper lacks discussion on relevant work in reasoning enhancement during the pre-training phase, such as https://arxiv.org/pdf/2404.07965.
> >
> > Thank you for noticing the connection between RHO-1 and our work. We have added a description of this paper to the end of our Related Works (Section 3, blue text). RHO-1 selectively trains on useful tokens during pre-training, which enhances reasoning downstream.  Also, after looking at RHO-1 in more detail, we’d be very excited for future work that combines RHO-1 with RA (i.e., to perform RL with CoT on datapoints that are suitable for reasoning, not noisy, and not yet learnt). We have included a mention of this at the end of Section 6.1 (blue text).
> >
> > We also commit to surveying the literature regarding other non-RL methods that enhance reasoning during pre-training and including a discussion of these works in the final copy of our manuscript.
> >
> > >The experiments are insufficiently comprehensive, as they are conducted on only one model and one dataset.
> >
> > We would like to emphasize that our paper contains multiple experiments which span multiple datasets: (1) our ”what reasoning to reward” experiment uses datapoints from FineWeb, (2) our “where reasoning is rewarded” experiment uses datapoints from MMLU, GSM8K, and CSQA, (3) our self-improving CoT reasoning experiment for the free-form QA setting evaluates on MMLU and GSM8K, and (4) our exploratory experiment on the grand challenge of the self-improvement on truly unstructured text uses OpenWebMath.
> >
> > That being said, we acknowledge that the paper would be strengthened by running the same experiments with different pretrained models of different sizes. Therefore, we have repeated the *what* and *where* experiments from Section 5.1 with another model, LLama 3.1 8B. We have added these results in Appendix B Table 5 and find them to be consistent with our original results using Mistral 7B in Table 2.
> >
> > >The presentation of the paper could be improved. Some key findings should be in the main body rather than the Appendix, such as Appendix D and the definition of RA in Appendix A. Essential parameters, like the type of LLM used and inference hyperparameters, should also be included in the main text.
> >
> > We thank the reviewer for identifying each of the specific ways in which clarity could be improved. We have reworked Appendix D into the new Section 6.2, and have added additional clarification to the definition of RA in Section 4. We’ve also added more information about the type of LLM used and inference hyperparameters (i.e., sampling temperature) in Section 5.1, Section 5.2, and Section 6.2 (in blue text).
> >
> >
> > We conclude by stressing that while we do not solve every challenge, our work represents a large step towards self-improving CoT reasoning on unstructured text at the pretraining scale. As more researchers from the LLM reasoning community shift focus towards this goal, we think that analyses like ours, which isolate and address specific issues with our current self-improvement methods, will provide great value and enable exciting future research.
> >
> > Thanks again for the thorough review. We hope that some of our additional explanations and experiments, along with the changes we've made in response to the issues you raised, go a long way towards improving the paper and changing your opinion. Please don’t hesitate to ask if you have any additional questions or require clarification.

---

> > > ### Comment · Reviewer_ZH4y · 2024-11-20
> > >
> > > Thanks for your reply, I improved my score accordingly.

---

> > > > ### Author Response · Authors · 2024-11-25
> > > >
> > > > Thank you for increasing your score and helpful initial review. We have since made a large number of improvements to the paper but, given that you are currently recommending that this paper not be accepted, are there any further improvements we could make that would change this?

---

> > > > > ### Comment · Reviewer_ZH4y · 2024-11-26
> > > > >
> > > > > Thank you for your detailed response and the significant improvements made to the paper. While your clarifications addressed some of my initial concerns, I still feel that the paper's contributions are somewhat limited. Specifically, while the analysis of reward functions is insightful, the work does not propose a sufficiently robust method to tackle the challenge of self-improving reasoning in LLMs comprehensively.
> > > > >
> > > > > I also remain uncertain about some aspects of the experimental results:
> > > > >
> > > > > The evaluation on MMLU is conducted under a zero-shot setting, whereas MMLU is more commonly assessed with 5-shot prompts. This makes it difficult to compare your results with standard baselines.
> > > > >
> > > > > In your response to Reviewer qdDb, you referenced Quiet-Star, which evaluated CSQA. Additionally, many recent works on enhancing LLM reasoning capabilities have used BBH for evaluation. Including both CSQA and BBH in your benchmarks would result in a more comprehensive evaluation.
> > > > >
> > > > > It remains unclear whether the proposed RA function will continue to provide benefits as model parameter counts increase. Would its effectiveness diminish as model performance improves?
> > > > >
> > > > > Given the substantial changes you’ve made compared to the initial submission (including changes to the title and contributions), I believe the paper could benefit from further refinement before being resubmitted in a future review cycle.

---

> > > > > > ### Author Response · Authors · 2024-11-27
> > > > > > **Response to Reviewer ZH4y**
> > > > > >
> > > > > > Thank you very much for your thoughtful feedback on our work. We appreciate the time and effort you’ve dedicated to reviewing our paper.
> > > > > >
> > > > > > >While your clarifications addressed some of my initial concerns, I still feel that the paper's contributions are somewhat limited. Specifically, while the analysis of reward functions is insightful, the work does not propose a sufficiently robust method to tackle the challenge of self-improving reasoning in LLMs comprehensively.
> > > > > >
> > > > > >
> > > > > > We acknowledge that our method does not yet achieve state-of-the-art results in self-improvement compared to existing supervised approaches. However, we would like to emphasize that the primary aim of our research was not necessarily to surpass current state-of-the-art methods in this domain. Instead, our work explores an exciting direction—self-improving reasoning without relying on supervised datasets. While the state of the art in this research direction currently yields less favorable empirical results than supervised methods, we believe that it could have significant long-term impact and is therefore worthy of continued attention.
> > > > > >
> > > > > > **Our study addresses a critical roadblock in this research direction**. We demonstrate that existing reward functions fall short, *even in an intermediate setting*, and introduce a novel reward function that succeeds where others fail. Moreover, we propose a new approach for evaluating reward functions: the "what/where" experiments, which help us identify the most effective reward function. We think that achieving complete self-improvement on unstructured text would be a groundbreaking result worthy of a very high (8-10) score. But given how many researchers—even in industry—are struggling with this challenge, systematic investigation of specific components like reward functions is crucial for advancing the field. We think this represents precisely the type of research progress that ICLR aims to promote.
> > > > > >
> > > > > > >The evaluation on MMLU is conducted under a zero-shot setting, whereas MMLU is more commonly assessed with 5-shot prompts. This makes it difficult to compare your results with standard baselines.
> > > > > > >
> > > > > > >In your response to Reviewer qdDb, you referenced Quiet-Star, which evaluated CSQA. Additionally, many recent works on enhancing LLM reasoning capabilities have used BBH for evaluation. Including both CSQA and BBH in your benchmarks would result in a more comprehensive evaluation.
> > > > > >
> > > > > > Thanks for the question. We base our evaluation methodology on the Quiet-STaR [1] paper, one of the most important works in LLM reasoning self-improvement. In this work, Zelikman et al. evaluate on two downstream reasoning benchmarks using zero-shot evaluation. We have taken the same approach in our work, but we’d be happy to include 5-shot prompting results in the final version of our paper.
> > > > > >
> > > > > > Moreover, while additional datasets would add to our evaluation, we believe that our current evaluation *over two models and five datasets* provides strong evidence for the strength of the RA reward function, especially considering that all the other reward functions fail to show any self-improvement. Note that our method should be compared to existing unsupervised methods, as opposed to methods using supervised datasets.
> > > > > >
> > > > > > >It remains unclear whether the proposed RA function will continue to provide benefits as model parameter counts increase. Would its effectiveness diminish as model performance improves?”
> > > > > >
> > > > > > Regarding model size, our experiments use 7B and 8B parameter models, which is standard practice in academic research. For example, Quiet-STaR [1] uses Mistral 7B, RAFT [2] uses Llama 7B, and RHO-1 [3] uses 1B and 7B models. As academic researchers, we show that our method works using academic computing resources. We believe this aligns with ICLR's academic focus, and we leave investigations using larger models to labs with industrial-scale compute budgets.
> > > > > >
> > > > > > Thanks again for your continued engagement with our work. We believe that the updates we have made in response to your comments and questions (red text) have significantly improved the quality and clarity of our paper.
> > > > > >
> > > > > > [1] Zelikman et al, Quiet-STaR: Language Models Can Teach Themselves to Think Before Speaking, 2024
> > > > > >
> > > > > > [2] Dong et al, RAFT: Reward rAnked FineTuning for Generative Foundation Model Alignment, 2023
> > > > > >
> > > > > > [3] Lin et al, Rho-1: Not All Tokens Are What You Need, 2024

---

### Official Review · Reviewer_qdDb · 2024-11-03

**Soundness:** 3
**Presentation:** 1
**Contribution:** 2
**Rating:** 5
**Confidence:** 4

**Summary:**

The paper titled explores the potential for self-improvement in large language models' ability to perform CoT reasoning without the need for supervised datasets. The authors frame this as a reinforcement learning problem where an LLM generates a CoT to predict subsequent tokens in a text corpus, receiving a reward based on the effectiveness of the CoT in predicting the next tokens. Their approach explores generating CoTs for next-token prediction in unstructured data, aiming to improve general-purpose reasoning abilities.

**Strengths:**

1. The paper presents a novel approach to improving CoT reasoning in LLMs, exploring reinforcement learning as a framework for unsupervised self-improvement. The introduction of RA offers an innovative solution to the reward function challenge.
2. This work addresses a crucial challenge in LLM development—achieving autonomous improvement in reasoning without reliance on human-generated data. If successful, this approach could significantly reduce reliance on expensive, curated datasets and enable more scalable reasoning improvement across diverse domains.

**Weaknesses:**

1. Some aspects of the reinforcement learning formulation could benefit from additional clarity, specifically regarding the choice of reward clipping values and the normalization strategies within RA. Additional explanation of these parameters and their impact on performance would make the approach more accessible.
2. The experiments focus primarily on a limited scope of problems (e.g., MMLU and OpenWebMath). The model’s performance on broader tasks, such as Tool learning or agent problem-solving scenarios, would offer stronger evidence of the approach’s generalizability.
3. By relying on the log-likelihood to evaluate the quality of intermediate reasoning (Chain-of-Thought) solely based on the model's ability to predict the following tokens, there is a risk that the model may overly focus on matching specific token patterns in the training data rather than developing generalized reasoning capabilities.

**Questions:**

See above.

---

> ### Author Response · Authors · 2024-11-19
> **Response to Reviewer qdDb**
>
> Thank you for your review and comments. We’re glad that you think our work addresses a crucial challenge in LLM development and that you see the introduction of RA as an innovative solution to the challenge of designing reward functions for reasoning. Please see below for responses to your comments and questions.
>
> >Some aspects of the reinforcement learning formulation could benefit from additional clarity, specifically regarding the choice of reward clipping values and the normalization strategies within RA. Additional explanation of these parameters and their impact on performance would make the approach more accessible.
>
> Thanks for the great suggestions. We have updated the parts of Section 4 discussing clipping and normalization to be more clear. We have also added a detailed discussion of their impact on performance to the end of Section 5.1 (the two big groups of blue text), including an additional ablation for different values of the clipping threshold. Moreover, Appendix B.1 contains tables which show full results for additional combinations of clipping, baseline, and normalization.
>
> Briefly summarizing how each design choice relevant to RA can be interpreted:
>
> - Clipping value: the minimum value at which suffix token log-probabilities are clamped. This prevents any given token from having an outsized loss contribution. We have run an additional ablation on a range of clipping values to demonstrate its impact, and include the results in Appendix B.1 (Figure 5).
>
> - Baseline: in RA, we compute the token log-probabilities for the suffix given the prefix and CoT, but subtract the "empty CoT" baseline, which is the token log-probabilities without conditioning on any CoTs (only the prefix). This ensures we are optimizing for CoTs that *improve* the suffix loss relative to not producing a CoT.
>
> - Normalization: normalizing by the "empty CoT" baseline re-scales the reward to ensure that we don't provide high reward values for CoTs with trivial-to-predict suffixes (ie, when an "empty CoT" predicts the suffix well).
>
> >The experiments focus primarily on a limited scope of problems (e.g., MMLU and OpenWebMath). The model’s performance on broader tasks, such as Tool learning or agent problem-solving scenarios, would offer stronger evidence of the approach’s generalizability.
>
> We thank the reviewer for their suggestion, it would indeed be interesting to evaluate the model’s tool learning and agentic behaviour. However, in keeping with much of the LLM reasoning literature, we chose to specifically focus our evaluations on a range of QA-style reasoning problems. One of the most important recent works on self-improving LLM reasoning, Quiet-STaR [1], uses two QA-style reasoning benchmarks to evaluate their method: GSM8K and CSQA. Similarly, we evaluate our self-improvement method from Section 5.2 on two QA-style reasoning benchmarks: MMLU and GSM8K.
>
> >By relying on the log-likelihood to evaluate the quality of intermediate reasoning (Chain-of-Thought) solely based on the model's ability to predict the following tokens, there is a risk that the model may overly focus on matching specific token patterns in the training data rather than developing generalized reasoning capabilities.
>
> We view the strong zero-shot transfer performance to GSM8K after optimising for RA on MMLU-FREE-FORM as compelling evidence that the model learns generalisable reasoning — beyond just matching specific token patterns in the data. Moreover, we strongly believe that RA being based on token loss is actually a key advantage. Since it is loss-based, it requires only one forward pass, does not require an external strong model, is not limited to exact-match heuristics like accuracy-based functions (which fail on unstructured text), and allows the model to place weight over a distribution of valid answers.
>
> It might also be worth mentioning that since we are starting from a pretrained LLM, much of the gains from attempting to match specific token patterns have already been exhausted. That is, at the beginning of standard LLM pretraining, most of the loss reduction is achieved by fitting specific token patterns like spelling and grammar rules, but by further reducing loss, models begin to learn higher-order skills such as CoT reasoning. We initialize our weights to a pretrained LLM, so the risk of overly focusing on specific token patterns is limited.
>
> Thanks again for your review and helpful suggestions. We are excited to see the insights from our work help progress the field towards the grand challenge of self-improving CoT reasoning on unstructured text at the pretraining scale. If you feel that we have adequately addressed your concerns, we would appreciate your consideration to increase our score. And if you have any additional questions or needs for clarification, please don’t hesitate to ask.
>
> [1] Zelikman et al, Quiet-STaR: Language Models Can Teach Themselves to Think Before Speaking, 2024

---

> > ### Comment · Reviewer_qdDb · 2024-11-22
> >
> > I really agree with ZH4y that the statement of pretraining stage may be misleading. I think the authors should carefully consider the contribution points of this paper, especially the term "pretraining". Meanwhile, the author should add more discussions on why the log-likelihood, as a reward signal, can promote the enhancement of the model's general reasoning ability.

---

> > > ### Author Response · Authors · 2024-11-22
> > > **Response to Reviewer qdDb**
> > >
> > > > I really agree with ZH4y that the statement of pretraining stage may be misleading. I think the authors should carefully consider the contribution points of this paper, especially the term "pretraining"
> > >
> > > Thank you for your response. We agree and have made considerable updates to the Introduction, Section 5, Section 6, and Conclusion with clearer descriptions of our contributions. The main updates are in *red text*. In our updates, we make it specifically clear that we do not solve the full unstructured pretraining setting.
> > >
> > > To make this even more clear, we also propose to change our title to “*On Reward Functions for Self-Improving CoT Reasoning Without Supervised Datasets*”.
> > >
> > > To our knowledge, our work is the first to provide this type of analysis on reward functions for self-improving CoT reasoning on unstructured text. We demonstrate that existing reward functions fail *even in* the simpler MMLU-FREE-FORM setting, introduce the novel RA reward function as an effective solution, and perform a comprehensive analysis of reward functions and how they affect what and where reasoning is rewarded. We believe that the successful zero-shot transfer from self-improving CoT reasoning on MMLU-FREE-FROM to a popular unseen math benchmark (GSM8K) is a promising result that motivates future work in this direction.
> > >
> > > Do our updates to the paper and the title change address your concerns regarding the clarity of our contributions?
> > >
> > > > Meanwhile, the author should add more discussions on why the log-likelihood, as a reward signal, can promote the enhancement of the model's general reasoning ability.
> > >
> > > Thank you for the suggestion. At the end Section 5.2 (*blue text*), we have added a discussion on how loss-based reward functions like RA can promote the enhancement of general reasoning ability. This discussion details how we reward reasoning that minimizes a form of loss on subsequent tokens, and references recent work [1] showing that optimizing for loss during pretraining improves performance on downstream reasoning tasks. We also explain how our experiments demonstrate that RA’s key modifications to standard loss (i.e., clipping, baseline, and normalization) are crucial for generalizing reasoning to unseen tasks.
> > >
> > > [1] Z. Du et al, Understanding Emergent Abilities of Language Models from the Loss Perspective, 2024

---

> > > > ### Comment · Reviewer_qdDb · 2024-11-24
> > > >
> > > > Thank you for your clarification and additional experiments. I will keep my score accordingly.

---

> > > > > ### Author Response · Authors · 2024-11-25
> > > > >
> > > > > Thank you again for providing an initial review that helped us to improve various aspects of the paper, both in terms of presentation and additional experiments. Having made extensive clarifications of the paper’s contributions and included additional discussion on why the log-likelihood is an appropriate starting point for a reward function for general-purpose reasoning, have we addressed all of your concerns? If so, will you consider increasing your score? Please do not hesitate to share any further questions.

---

### Official Review · Reviewer_KyrV · 2024-11-04

**Soundness:** 3
**Presentation:** 2
**Contribution:** 3
**Rating:** 6
**Confidence:** 3

**Summary:**

This paper explores how to enable large language models (LLMs) to self-improve their Chain-of-Thought (CoT) reasoning abilities using general pre-training data rather than supervised datasets. The authors investigate what makes a good reward function for learning reasoning during language modeling, examining how different reward functions affect both what reasoning is rewarded and where reasoning is applied. They introduce a novel "Reasoning Advantage (RA)" reward function that combines clipping and normalization techniques, and demonstrate its effectiveness on a new free-form question-answering dataset called MMLU-FREE-FORM, showing improved transfer to math reasoning tasks.

**Strengths:**

The systematic analysis of reward functions and their properties is thorough and well-motivated. The introduction of the RA reward function addresses key limitations of existing approaches, particularly in distinguishing good reasoning from random text and identifying appropriate contexts for reasoning. The creation of MMLU-FREE-FORM as an intermediate benchmark between structured QA and general language modeling is clever and useful for the research community. The empirical results showing successful transfer learning to GSM8K math problems provide concrete validation of their approach.

**Weaknesses:**

The paper's primary limitation appears in the scaling to general pre-training data, where the offline reinforcement learning approach that worked well on MMLU-FREE-FORM struggles to escape local optima of conservative reasoning. While the authors acknowledge this limitation and suggest future research directions, the paper doesn't fully solve the challenge of self-improving reasoning at pre-training scale. Additionally, while the authors demonstrate improved performance on mathematical reasoning tasks, there could be more exploration of how well their approach generalizes to other types of reasoning beyond mathematics.

**Questions:**

Have you explored whether the effectiveness of Reasoning Advantage (RA) varies across different types of reasoning tasks beyond math and standard QA?
In Section 5.2, you show that optimizing for RA leads to a 7% improvement on GSM8K. Could you provide more analysis of what specifically improved in the model's reasoning capabilities? Are there particular types of math problems where the improvement was more pronounced?
The paper mentions that only 0.01% of generated CoTs achieve a reward above 0.2 on OpenWebMath. Have you analyzed these high-scoring CoTs to understand what makes them successful? This analysis could inform better prompting strategies.

---

> ### Author Response · Authors · 2024-11-19
> **Response to Reviewer KyrV [1/2]**
>
> We thank the reviewer for the thoughtful review, and for recognizing the value in using MMLU-FREE-FORM as an intermediate benchmark between curated QA data and the unstructured text setting. We address your questions and comments below.
>
> >The paper's primary limitation appears in the scaling to general pre-training data, where the offline reinforcement learning approach that worked well on MMLU-FREE-FORM struggles to escape local optima of conservative reasoning. While the authors acknowledge this limitation and suggest future research directions, the paper doesn't fully solve the challenge of self-improving reasoning at pre-training scale.
>
> We agree with the reviewer that our work does not solve the grand challenge of self-improving CoT reasoning on unstructured, pretraining-scale text. Our primary contributions are showing that standard reward functions fail *even in* the intermediate MMLU-FREE-FORM setting, introducing a novel reward function to solve this issue, and performing a comprehensive analysis of reward functions and how they affect *what* and *where* reasoning is rewarded. To our knowledge, our work is the first to provide this type of analysis on reward functions for self-improving CoT reasoning on unstructured text.
>
> We also perform an exploratory experiment on the full unstructured setting and provide key insights to help facilitate future research in this direction. We believe these to be major contributions to the literature, and have dramatically updated our Introduction, Section 5, Section 6, and Conclusion with more targeted descriptions of our contributions. The key points are updated with red text.
>
> >While the authors demonstrate improved performance on mathematical reasoning tasks, there could be more exploration of how well their approach generalizes to other types of reasoning beyond mathematics.
>
> We thank the reviewer for this suggestion. While GSM8K focuses purely on mathematics, MMLU contains questions which span various fields, including mathematics, sciences, law, etc. On MMLU, we actually find that our method leads to large gains on questions across a wide span of subjects (biology, physics, accounting, law, computer science, etc.) that involve quantitative reasoning—*going beyond just mathematics*. We have updated Appendix B.2 to better explain this point (blue highlighted text). Thanks again for the suggestion.

---

> ### Author Response · Authors · 2024-11-19
> **Response to Reviewer KyrV [2/2]**
>
> >Have you explored whether the effectiveness of Reasoning Advantage (RA) varies across different types of reasoning tasks beyond math and standard QA?
>
> In keeping with much of the LLM reasoning literature, we chose to specifically focus our evaluations on a range of QA-style reasoning problems. However, we do show positive results that go beyond just mathematics. As mentioned above, Appendix B.2 Figure 6 shows that RA significantly improves reasoning performance on “reasoning style questions” in the MMLU test set. These questions span a wide range of subjects beyond math, including physics, biology, accounting, law, computer science, etc.
>
> >In Section 5.2, you show that optimizing for RA leads to a 7% improvement on GSM8K. Could you provide more analysis of what specifically improved in the model's reasoning capabilities? Are there particular types of math problems where the improvement was more pronounced?
>
> Great question. We observe that the GSM8K accuracy goes up due to fewer logical/arithmetic errors in the generated CoTs, but we don’t observe a single predominant qualitative change. However, we do notice something interesting regarding performance on the MMLU test set. As mentioned above, MMLU sees an improvement in questions across a wide range of subjects that require quantitative reasoning (biology, physics, accounting, computer science, etc.). Moreover, the improvement is far larger for these questions than for those that require recall (see Figure 6 in Appendix B.2). Thinking about it, this result makes a lot of sense, since CoT reasoning probably doesn't help as much when trying to recall a fact. However, for quantitative reasoning and problem-solving tasks, additional reasoning can clearly be of benefit. We have added a discussion of this to Appendix B.2 (blue text). Thank you for the suggestion!
>
> >The paper mentions that only 0.01% of generated CoTs achieve a reward above 0.2 on OpenWebMath. Have you analyzed these high-scoring CoTs to understand what makes them successful? This analysis could inform better prompting strategies.
>
> Thanks for asking this great question. Upon manual inspection, many of the CoTs that passed the filtering threshold exhibited the conservative strategy described in the paper: they simply summarize past information from the context. This explains why the model learned to be overly conservative. However, these overly conservative CoTs which made it past the RA threshold were still superior to those that did not pass the threshold (the ones that did not pass the threshold mainly contained **incorrect** reasoning that predicted the subsequent tokens incorrectly). This indicates that RA actually succeeded at its job of identifying the best reasoning from the generated batch of CoTs, and that the main issue indeed lies with the *lack of diversity* in the generated CoTs. We mention a few potential ways to generate more diverse CoTs in the paper, but we agree that it would also be worth exploring different prompting strategies (we used a single system prompt to generate these CoTs, and did not spend much time on prompt engineering).
>
> We have added this detailed discussion to Section 6.1 (the first blue block text, and then the rest discusses ways to increase diversity), and we think it dramatically improves the section. Thanks again for the great suggestion.

---

> > ### Comment · Reviewer_KyrV · 2024-11-24
> >
> > Thanks for the response, I will keep with my original score

---

> > > ### Author Response · Authors · 2024-11-25
> > >
> > > Thank you again for helping to improve our paper with the suggestions and comments in your initial review. In light of having provided points of clarification, updated the paper accordingly, and received no further questions, we ask if you would consider increasing the confidence in your score. And if you still have any further questions, please share those with us so that we can ensure the clarity of our work.

---

### Author Response · Authors · 2024-11-19
**General Response From Authors to Reviewers**

Thanks to all the reviewers for your time and effort during the review process. We appreciate that you found our work insightful, and we’re glad that there is excitement about our progress towards self-improving CoT reasoning. Your thoughtful reviews have helped us dramatically improve the clarity and rigour of our submission.

We have responded to each reviewer individually, and also include a general response to all reviewers here. If you find our answers responsive to your concerns, we would be grateful if you considered increasing your score, and if you have additional questions, we’re happy to engage further.


>Additional experiments.

Based on the reviewers’ questions and comments, we have added further experiments to strengthen our results. In response to reviewers ZH4y and MbaU, we have added results going beyond a single model to also include Llama 3.1 8B (see Table 5 in the appendix). In response to reviewer QdDb, we have included an additional ablation for different values of the clipping threshold (Figure 5 in the appendix). And in response to reviewer KyrV, we have added a qualitative analysis of the highest- and lowest-scoring CoTs in Section 6.1.

>Clarification of contributions.

We have updated the submission pdf to clarify the contributions of our work (main updates in red text). A brief summary is provided below:

As it becomes increasingly challenging and prohibitively expensive to curate large-scale (question, CoT, answer) datasets, the LLM reasoning community has begun focusing on a grand challenge task: self-improving CoT reasoning on unstructured, pretraining-scale text. To be clear, our work does not solve this grand challenge. Our primary contributions are showing that standard reward functions fail *even in* the intermediate MMLU-FREE-FORM setting, introducing a novel reward function to solve this issue, and performing a comprehensive analysis of reward functions and how they affect *what* and *where* reasoning is rewarded. To our knowledge, our work is the first to provide this type of analysis on reward functions for self-improving CoT reasoning on unstructured text. There is still more work to be done in order to solve the full unstructured pretraining setting, and we present an exploratory experiment that provides key insights to help facilitate future research in this direction. We believe these to be major contributions to the literature, and have dramatically updated our manuscript (especially the Introduction, Section 5, Section 6, and Conclusion) with more targeted and clear descriptions of our contributions.

Note: The main updates to the manuscript which clarify our contributions are made with *red* text. The blue text refers to changes made in response to specific reviewer questions, and we reference them in the individual responses below.

>Updated appendix.

We have also updated the appendix to be more clear, and have reworked the previous Appendix D into the main paper as the newly added Section 6.2 (per the request from reviewer ZH4y). To make space for these changes we have:
- Converted the former Figure 3 bar plot into a table (Table 2). Notice that full results with confidence bounds can still be found in Appendix B.1.
- Moved the former Figure 1 to the Appendix (new Figure 7).
- Moved the former Figure 5 to the Appendix (new Figure 8).

>Clarified hyperparameters.

We have made the specific details about model architecture and inference hyperparameters (i.e., temperature) more clear in the manuscript. These changes are made in blue text.


We again thank the reviewers for their engagement and we appreciate all the suggestions that we believe will make the paper significantly stronger!

---

### Meta-Review · Area_Chair_DYu9 · 2024-12-22

**Metareview:**

The paper introduces the Reasoning Advantage (RA) reward function to improve CoT reasoning in LLMs using unstructured, pretraining-scale text instead of curated datasets. RA demonstrates improved performance on MMLU-FREE-FORM and transferability to GSM8K math tasks, while also providing a detailed analysis of reward functions and their properties. The work addresses a critical challenge in LLM reasoning, introducing MMLU-FREE-FORM as an intermediate benchmark that bridges curated QA datasets and unstructured text. The experiments highlight RA’s ability to improve reasoning performance and generalization to diverse tasks. However, some reviewers felt the contributions were incremental, with RA being a slight modification of token loss and MMLU-FREE-FORM a minor adaptation of MMLU. They also noted the limited scope of models and datasets tested, along with minimal improvement on OpenWebMath, raising concerns about scalability to pretraining-scale data. While the approach shows promise in intermediate settings, the lack of comprehensive evaluations and broader generalizability testing would limit its impact. Future work might expand experiments to include larger and more diverse datasets (e.g., BBH, CSQA) and evaluate performance across multiple model sizes to validate scalability and robustness.

**Additional Comments On Reviewer Discussion:**

While the authors made significant efforts to address the reviewers' concerns by providing clarifications, adding experiments, and improving the paper’s presentation, the reviewers maintained their original scores. Despite acknowledging the improved clarity and additional contributions, they felt that the work's technical novelty and scalability remain limited, suggesting the paper could benefit from further refinement and resubmission in the future.

---

### Decision · Program_Chairs · 2025-01-22

Reject